# **Evaluation of the Dual Gamma Generalized Extreme Value Distribution for flood events in Poland**

Łukasz Gruss<sup>1</sup>, Patrick Willems<sup>2</sup>, Paweł Tomczyk<sup>1</sup>, Jaroslav Pollert Jr. <sup>3</sup>, Jaroslav Pollert Sr. <sup>3</sup>, Christoph Märtner<sup>4</sup>, Stanisław Czaban<sup>1</sup> and Mirosław Wiatkowski<sup>1</sup>

Institute of Environmental Engineering, Wrocław University of Environmental and Life Sciences, Wrocław, 50-363, Poland Department of Civil Engineering, KU Leuven, Kasteelpark Arenberg 40, 3001, Leuven, Belgium

<sup>3</sup>Faculty of Civil Engineering, Czech Technical University in Prague, Prague, 16629, Czech Republic

Correspondence to: Łukasz Gruss (lukasz.gruss@upwr.edu.pl)

# 10 Abbreviations and acronyms in alphabetical order

- a micro-catchments
- b meso-catchments
- c macro-catchments
- d large catchments
- e very large catchments
  - A catchment area [km2]
  - CRPS Continuous Ranked Probability Score
  - Exp exponential distribution
  - FFA Flood frequency analysis
- GEV generalized extreme value distribution
  - GGEV Dual Gamma Generalized Extreme Value Distribution
  - GOF Goodness of fit test
  - H highland
  - Kurt. kurtosis
- L lowland
  - LN2 2-parameter log-normal distribution
  - LN3 3-parameter log-normal distribution
  - M mountain
  - $MAE-Mean\ Absolute\ Error$
- MAF mean annual maximum flow
  - $MK-The\ Mann-Kendall\ trend\ test$
  - MLE maximum likelihood estimation
  - $N-sample \ size$
  - $NMT-no\ trend$
- NT negative trend
  - P3 Pearson type III distribution
  - PT positive trend
  - Qp Peak flow
  - RDA Redundancy Analysis
- RMSE Root mean square error
  - SD standard deviation

<sup>&</sup>lt;sup>4</sup>M&S Umweltprojekt GmbH, Pfortenstraße 7, 08527 Plauen, Germany

Skew. - empirical skewness

Var – Variance

VIF - Variance Inflation Factor

**Abstract.** Climate change has already affected global water resources and is expected to have even more severe consequences in the future. Advancing climate change will necessitate the use of new distributions that are more flexible in adapting to trends and other non-stationarities. In this paper we compare three-parameter distributions such as the log-normal, the Generalized Extreme Value (GEV) and the Pearson type III with the Dual Gamma Generalized Extreme Value Distribution (GGEV). The GGEV is a four-parameter extension of the GEV. The comparison is done under different trend conditions and takes into 50 account the differences in the catchment area and peak flow magnitude. The research pertains to basins in the temperate climate zone of Poland and includes data from 678 water gauges located on 340 rivers. Based on the trend criterion, the GGEV distribution compared to the analyzed three-parameter distributions, and the GEV distribution compared to the other threeparameter distributions, were the best fit for most samples. Based on the trend criterion and the catchment size, GEV is best suited for micro- and meso-catchments, while GGEV is ideal for macro- to large-catchments when the series exhibits a trend, either positive or negative. The major benefit of GGEV is its flexibility when the data are influenced by temporal non-55 stationarities. The additional shape parameter of GGEV compensates for the limitations of the other shape parameter in distributions with lighter tails. Analysis of the dependence relationships between the environmental indicators such as the geographic, physiographic and hydrological indicators as well as the distribution parameters is less conclusive. In order to test the risk of overparameterization and overfitting for the distributions with more parameters, the Kolmogorov-Smirnov test and the K-Fold cross validation were used. They show that the GEV and GGEV distributions perform better compared to the exponential and the two-parameter lognormal distributions. As an overall conclusion, the study shows that for the analyzed samples from the temperate climate zone in the era of climate change, distributions that better capture trends – such as GGEV perform more effectively.

#### 1 Introduction

80

Climate change has already affected global water resources and is expected to have even more severe consequences in the future (Dakhlaoui et al., 2019; Pokhrel et al., 2021; Połomski and Wiatkowski, 2023; Tomczyk et al., 2023; Willems, 2013). The significance of climate change lies in the substantial impacts it brings, including the increased frequency of floods (Gruss et al., 2023; Tabari et al., 2021b). In modeling of extreme hydrological events, such as floods, stochastic modeling is commonly used. This approach relies on historical data and employs probability distributions (Gruss et al., 2022; Młyński et al., 2020) to account for the uncertainty and variability of these phenomena (Szulczewski and Jakubowski, 2018). Such methods include the at-site flood frequency analysis (FFA) (Cassalho et al., 2018). The choice of probability distribution should be verified against the assumptions of stationarity and independence, as any deviation may result in biased outcomes and potentially catastrophic consequences, such as inappropriate designs, that could endanger property and human life (Ologhadien, 2021). However, the assumption of stationarity has faced increasing challenges due to the intensification of climate change and human activities (Gruss et al., 2022; Jiang and Kang, 2019; Milly et al., 2008). Many studies present series consisting of annual maximums where, for some water gauges, the assumption of stationarity, randomness, or non-monotonic trend (*NMT*) is not met (Cassalho et al., 2018; Szulczewski and Jakubowski, 2018). Advancing climate change will require the use of new distributions that are more flexible in adapting to changes in stationarity or the presence of trends in the sample.

In many countries, two- and three-parameter distributions are used to estimate the magnitude and frequency of annual maximum streamflow (e.g. Valentini et al., 2024; Gruss et al., 2022; Młyński et al., 2018; Pitlick, 1994; Rutkowska et al., 2015; Rutkowska et al., 2015; Bezak et al., 2014; Morlot et al., 2019; Šraj et al., 2016; Ul Hassan et al., 2019; Berton and Rahmani, 2024). There are also many studies among them on the Pearson type III (P3) (Cassalho et al., 2018), the log Pearson

Type III (LP3) (Berton and Rahmani, 2024; Morlot et al., 2019) and the two- and three-parameter log-normal distributions (LN2 and LN3) in the at-site FFA (Cassalho et al., 2018). According to Vogel and Wilson (1996), the LP3 provides the best fit to both annual minimum and the annual average streamflows, assuming the series is stationary. In recent decades, a significant amount of research has been dedicated to the Generalized Extreme Value distribution (GEV). Extreme events are often better modeled using heavy tailed distributions (Karczewski et al., 2022; Karczewski and Michalski, 2022), a characteristic of the GEV distribution (Cassalho et al., 2018; Morlot et al., 2019; Otiniano et al., 2019; Rutkowska et al., 2015). However, some extreme event data do not follow the GEV distribution because they require a more asymmetric distribution or one with a heavier tail. As a result, new classes of probability distributions have been developed that extend beyond the GEV, such as the Dual Gamma GEV distribution (GGEV) (Otiniano et al., 2019). The GGEV distribution is regarded as highly flexible for several reasons: 1. It introduces an additional parameter that adjusts tail weight and skewness, making it more adaptable to diverse datasets. 2. This added flexibility allows the GGEV to capture the nuances of empirical data more effectively than the standard GEV. 3. Consequently, the GGEV distribution is often preferred in practical applications where accurate modeling of complex data is essential (Nascimento et al., 2015). The additional shape parameter ( $\delta$ ) enables the GGEV distribution to adapt to various data characteristics, especially in terms of tail behavior. Notably, when this parameter is less than 1, the GGEV exhibits a heavier tail than the GEV, making it more effective at modeling extreme events that may occur more frequently than lighter-tailed distributions would predict (Silva and Do Nascimento, 2022).

Next to the influence of non-stationarities, it is well-known that various environmental factors, including land use, may significantly influence the tail of flood frequency distributions, although this depends on the region. Pitlick (1994) has found that the mean annual flood is most closely correlated with the watershed area, but did not find an influence of other measures of basin physiography on the differences in flood frequency distributions. In contrast, research by Ahilan et al. (2012) confirms that the type of landscape influences the GEV. Other research by Sampaio and Costa (2021) and Tyralis et al. (2019) has shown that morphological catchment characteristics correlate with these distributions. Also Kusumastuti (2007) highlights the role of environmental factors in influencing flood frequency and the occurrence of flood events. Although single factors may not always correlate well with the distribution parameters, it may be the combined influence of multiple factors that explain the differences in flood quantiles (Allamano et al., 2009). Understanding this influence may provide valuable insights for regionalization (He et al., 2015) and reduce uncertainties in inferences made using regional FFA frameworks (Hu et al., 2020; Tyralis et al., 2019). In this study, the assumption is made that if environmental factors have an influence on the distribution parameters, one can expect dependence relationships between the parameters when different distributions are calibrated to the flood data.

The aim of the study is to analyze the fit of the GGEV distribution versus the three-parameter distributions (GEV, LN3, P3) to empirical data for river basins in Poland. The study also aims to analyze the consistency of patterns exhibited by environmental factors with regard to the parameters of the examined distributions and to conduct tests for overparameterization and overfitting of the analyzed distributions.

## 2 Study area








The research area spans 678 water gauges situated within the drainage basins of the Dniester, Dunajec, Neman, Oder, Pregoła, Vistula, and other rivers flowing into the Baltic Sea and covering the territory of Poland in Central Europe (Figure 1). Poland is located within the temperate climate zone.

Figure 1. Location of the analyzed 678 water gauges (Source: hydrographic map of Poland).



Depending on the size of the catchment area in the study area, micro-catchments ( $A < 10 \text{ km}^2$ ), meso-catchments ( $10 \le A < 100 \text{ km}^2$ ), macro-catchments ( $100 \le A < 1,000 \text{ km}^2$ ), large catchments ( $1,000 \le A < 10,000 \text{ km}^2$ ), and very large catchments ( $A > 10,000 \text{ km}^2$ ) were distinguished. This division criterion is adopted based on Bertola et al. (2020). The fewest catchments identified were micro-catchments, represented by only 2 stream gauge profiles, while the most numerous were macro-catchments, represented by 388 profiles. In between were the very large catchments, meso-catchments, and large catchments, with 50, 68, and 170 stream gauge profiles, respectively (Fig. 1).

Figure 2. Terrain characteristics of the analyzed 678 stream gauge profiles consist of lowlands, highlands, and mountains (Source: hydrographic map of Poland).

The terrain of the study area is uneven. Most stream gauge profiles (as many as 582 profiles) are located in lowland areas (Fig. 2). These are catchments situated within the provinces of the Central Polish Lowland, Eastern Baltic-Belarusian Lowland, and the Czech Massif, on the Polish Uplands. A smaller quantity, specifically 86 stream gauge profiles, were located in highland areas. They are located within the provinces of the Polish Uplands, Czech Massif, and the Western Carpathians with the Western and Northern Subcarpathia, as well as the Eastern Carpathians with the Eastern Subcarpathia. A total of ten stream gauge profiles located in mountainous regions are situated within the provinces of the Czech Massif and the Western Carpathians with the Western and Northern Subcarpathia, as well as the Eastern Carpathians with the Eastern Subcarpathia.

### 3 Methods



## 3.1 Data collection and extraction of flow extremes

For 1070 gauge stations located in the basins of the Vistula, Oder, coastal rivers, Pregoła, and Neman, Dniester, Dunajec, the maximum annual flows were collected. The source of the data (flows) is the Institute of Meteorology and Water Management - National Research Institute (IMGW-PIB). These data have been processed. Only the gauge stations with data series equal to or longer than 30 years were retained (Gruss et al., 2022; Tabari et al., 2021a). The data periods used for analysis varied across stations, from 30 to 70. In this way, the maximum annual flows were collected for 678 stations. These data were compiled in the hydrological year, which for Poland begins in November and ends in October. For each hydrological year, the annual maximum flow was extracted. These values hereafter referred to as peak flows (*Qp*), often associated with floods or extreme hydrological events (Gruss et al., 2022; Langridge et al., 2020; Northrop, 2004). *Qp* help in understanding the maximum capacity of rivers or streams to handle water, which is essential for infrastructure planning, floodplain management, and disaster mitigation efforts (Langridge et al., 2020). The *Qp* were utilized in this study for calibrating and evaluating the probability distributions.

For each station, the mean annual flood or mean annual maximum flow (*MAF*) represents the average of the *Qp* over the period of record (Nyeko-Ogiramoi et al., 2012; Pastor et al., 2014). In order to understand the long-term characteristics of river systems, including flood frequency, river behavior, and water resource management, the hydrologists often analyze mean annual maximum flows (Merz and Blöschl, 2009; Nyeko-Ogiramoi et al., 2012; Pastor et al., 2014). In this study, the *MAF* was utilized for the redundancy analysis.

#### 3.2 Trend detection


For all the analyzed time series, a test was conducted to ascertain the presence of a trend. The Mann-Kendall test was utilized for this purpose. This allowed us to group the obtained distributions into three categories: without a trend, with a positive trend, and with a negative trend.

The Mann-Kendall test (MK) is frequently used to detect a monotonic trend in long time series of hydrological data (Cassalho et al., 2018; Gruss et al., 2022, 2023; Rutkowska, 2015; Svensson et al., 2005).

The null hypothesis is that the data are identically distributed, the alternative hypothesis is that the data follow a monotonic trend. A two-sided test was performed and the significance level was set at 5%.

## 3.3 Extreme value distributions and parameter calibration methods

This study considered the following type of extreme value distributions: the four-parameter GGEV distribution and the three-parameter distributions GEV, LN3, and P3.

The GGEV probability density function (PDF) proposed by (Nascimento et al., 2015), is given by:

$$f(x; \mu; \sigma; \xi; \delta) = \begin{cases} \frac{\sigma^{-1}}{\Gamma(\delta)} \left[ 1 + \frac{\xi(x-\mu)}{\sigma} \right]^{-\left(\frac{\delta}{\xi}\right) - 1} \exp\left\{ -\left[ 1 + \frac{\xi(x-\mu)}{\sigma} \right]^{-\frac{1}{\xi}} \right\}, \xi \neq 0 \\ \frac{\sigma^{-1}}{\Gamma(\delta)} exp\{ -\delta[(x-\mu)/\sigma] \} exp\left\{ -exp\left\{ \left[ -\frac{x-\mu}{\sigma} \right] \right\}, \xi \to 0 \end{cases}$$
 (1)

where:

 $\mu$  – location parameter

 $\sigma$  – scale parameter

 $\xi$  – shape parameter

 $\delta$  – shape parameter of the GGEV extension.

This GGEV is a four-parameter extension of the GEV distribution with an additional shape parameter (δ). The three parameters of this distribution: μ, σ, ξ were estimated a posteriori by the ggevp function from the MCMC4Extremes R package (in this study: block = 1, int = 100; computer: AMD Ryzen 7 4800H with Radeon Graphics 2.90 GHz, RAM: 16 GB; SSD: 500 GB and 1 TB) (Do Nascimento and Moura E Silva, 2015; R Core Team, 2022). The initial 33,333.3 iterations – corresponding to the first portion of the chain (first thin\*int/3 iterations) – were designated as the burn-in period. The Metropolis-Hastings algorithm technique of sampling was used to estimate the marginal posterior distribution for each parameter except for δ, because identifiability problems were detected in the estimation of this fourth parameter, as reported by (Nascimento et al., 2015). For the estimation of the δ parameter, the method proposed by Nascimento et al. (2015) and (Silva and Do Nascimento, 2022) was adopted. We created a grid of possible δ values from 0.01 to 10, with an increment of 0.01, to estimate the other parameters (μ, σ, ξ) using the Bayesian approach for each point of the grid and selected the δ value with the lowest -2ln(L).

The calibration of the fourth parameter δ is thus not based on a purely Bayesian approach, but limited to the capabilities of the ggevp function (Silva and Do Nascimento, 2022).

The posteriori parameter values were selected by the lowest AIC, BIC and DIC values (Belzile et al., 2023).

The diagnosis was for each chain based on trace plots using the plot.mcmc function from the coda R package (Plummer et al., 1999). We made trace plots separately for each of the three parameters ( $\mu$ ,  $\sigma$ , and  $\xi$ ). For the  $\mu$  parameter, in each profile (in

678 profiles), the chain reached stationarity (the values of  $\mu$  oscillate around the mean for most of the iterations). For the  $\xi$  parameter in 8 profiles, and for  $\sigma$  in 58 profiles, the chain did not reach stationarity. When analyzing the plot with three chains, it can be seen that their courses are quite consistent for all of the three analyzed parameters (they overlap and do not separate clearly from each other, which suggests their convergence).

Moreover, to assess the convergence of the model, the scale reduction factor  $\widehat{R}$  (Gelman and Rubin's convergence diagnostic) was run for three parallel chains (Hamra et al., 2013; Ossandón et al., 2022). For this purpose, the gelman.diag function from the coda R package was used. The  $\widehat{R}$  values below the critical threshold of 1.1 indicate adequate model convergence. In all our runs, the  $\widehat{R}$  values were below 1.1 in 678 samples, confirming convergence. The  $\widehat{R}$  factor is commonly seen as a convergence diagnostic, useful for finding sufficient burn-in (Jones and Qin, 2022; Vats and Knudson, 2021).

The GEV was used in many studies (Abida and Ellouze, 2008; Bezak et al., 2014; Cassalho et al., 2018; Kidson and Richards, 2005; Szulczewski and Jakubowski, 2018). The GEV PDF function is given in equation (2):

$$f(x) = \exp[-\{1 + \frac{\xi(x-\mu)}{\sigma}\}^{-1/\xi}],\tag{2}$$

for  $1+\xi(x-\mu)/\sigma>0$ , and  $\sigma>0$ , where:

 $\mu$ ,  $\sigma$ ,  $\xi$  are location, scale and shape parameters, respectively.

The parameters of this distribution were estimated by the maximum likelihood method (MLE), as described by Smith (1985). The estimation of the parameters and fitting of the GEV distribution was done using the 'evd' and 'fExtremes' R packages (Stephenson, 2024; Wuertz et al., 2023).

The LN3 distribution function is given by the formula (3):

$$f(x) = \frac{1}{(x-\mu)\sigma_{yy/2\pi}} \exp\{-\frac{1}{2\sigma_y^2} [\log(x-\mu) - \xi_y]^2\},\tag{3}$$

where:

205

 $\xi_{\rm y},\sigma_{\rm y}^2$ ,  $\mu$  are shape, scale and location, parameters, respectively.

The LN3 is similar to the two-parameter LN2 distribution, except that x is subtracted by a value α in the former, which represents the lower bound (Cassalho et al., 2018). The parameters of this distribution were estimated by MLE as shown by Meeker and Escobar (1998). The estimation of the parameters and fitting of probability distribution was carried out using the: 'EnvStats' and 'weibulltools' R packages (Hensel and Barkemeyer, 2018; Millard, 2013). For the MLE method used to estimate the distribution parameters a confidence level of 0.95 was assumed.

220 The PDF of the P3 distribution is given by (4):

$$f(x) = \frac{1}{|\sigma|^{\xi} \Gamma(\xi)} |x - \mu|^{\xi - 1} e^{-\frac{x - \mu}{\sigma}},$$
for  $s \neq 0$ ,  $a > 0$  and  $\frac{x - \lambda}{s} \geq 0$ . (4)

Where:

 $\xi$ ,  $\sigma$ ,  $\mu$  are shape, scale and location parameters, respectively.

The MLE was used to estimate the parameters for the P3 distribution. In the gamma distribution developed by Becker and Klößner (2025), this function allows for negative scale parameters to accommodate negative skewness. The estimation of the parameters and fitting of probability distribution was done using the 'PearsonDS' R package (Becker and Klößner, 2025).

### 3.4 Accuracy measures and scoring rules

The goodness of fit of the four probability distributions to the empirical data was evaluated based on the accuracy measures

Mean Absolute Error (MAE) and Root mean square error (RMSE). The MAE is recommended for leptokurtic distributions

(MAE) and RMSE is preferred for platykurtic distributions (Karunasingha, 2022). Among the 678 samples, the kurtosis value exceeded 3 for 560 samples (leptokurtic distributions), while kurtosis less than 3 was observed in 118 samples (platykurtic distributions). Moreover, the Continuous Ranked Probability Score (CRPS) was used to compare the entire CDF distribution (Hersbach, 2000; Pic et al., 2025). For this purpose, the crps\_sample function from the scoringRules R package was used

(Jordan et al., 2016).

## 3.5 Redundancy analysis

Redundancy analysis (RDA) was applied as a canonical technique, to investigate the influence of environmental variables and sample characteristics on the parameters of the extreme value distributions. The aim was to identify common patterns and key factors affecting the distribution parameters.

- The environmental factors examined included the watershed area, categorized by the catchment type, and the nature of the watercourse (Lowlands, Highlands, Mountains) (Bertola et al., 2020; Han et al., 2023; Tyralis et al., 2019). Sample characteristics considered included the highest *Qp*, *MAF*, sample size (*N*), empirical moments of standard deviation (*SD*), variance (*Var*), skewness (*Skew*.), kurtosis (*Kurt*.), third-moment center (3thMoment), fourth-moment center (*4thMoment*; which measures the intensity of the distribution tails) of the *Qp*, and trend measures.
- RDA was performed separately for each distribution. The final RDA model was selected by evaluating independent variables using the Variance Inflation Factor (VIF). The independent variables of the RDA are catchment area ranges: a microcatchments, b meso-catchments, c macro-catchments, d large catchments, e very large catchments; Qp; trend (NMT is no trend, PT is positive trend and NT is negative trend); the nature of the watercourse (L lowlands, H highlands, M mountains); Skew, AthMoment; Kurt., and N. The response variables of the RDA are  $\delta$ ,  $\mu$ ,  $\sigma$  and  $\xi$ .
- Since the *mean*, *SD*, *Var*, *3thMoment*, and *4thMoment* are interrelated, it is essential to carefully select the set of explanatory variables. It was also confirmed that multicollinearity exists between *Skew*. and *Kurt*. as explanatory variables (VIF > 10). Collinearity between *Skew*. and *Kurt*. may result from the fact that both of these measures are defined using the *SD*. Therefore, RDA was conducted separately for *Kurt*. (Fig. 9a, 10a, 11a, 12a) and *Skew*. (Fig. 9b, 10b, 11b, 12b). Additionally, RDA was performed with the inclusion of the catchment area ranges and *Kurt*. was replaced with *4thMoment* and *Skew*. (see Figures 9c, 10c, 11c, and 12c).
  - The decision to replace *Kurt*. with *4thMoment* was made because both *Skew*. and *Kurt*. are functions of *SD*, making them potentially collinear. The use of *Skew*. and *4thMoment* allows for capturing more detailed aspects of the data distribution. *4thMoment* measures the overall *Kurt*., which is the tail heaviness of the distribution, while *Kurt*. is the normalized version of this moment. Following the initial RDA, subsequent analyses considered only the changes that were not identified in the first analysis.

The use of topography in modeling Qp helps to uncover the runoff mechanism prevailing in the catchment (Valeo and Rasmussen, 2000).

RDA, standardized by response variables (center and standardize) and environmental variables (center), was performed using the Canoco software ver. 5.12 (ter Braak and Šmilauer, 2019).

## 265 3.6. Assessment of overparameterization and overfitting


Increasing the number of parameters in a distribution does not automatically improve its accuracy. Clearly, it will lead to a better goodness-of-fit, because of the higher flexibility during calibration, but there being more parameters will also lead to a higher uncertainty in their calibration. Consequently, a three- or more parameter distribution may result in

"overparameterization" and "overfitting". Including more parameters also increases the risk of greater errors in distribution extrapolations (Alsadat et al., 2023).







In order to evaluate whether the increased complexity of multi-parameter distributions offers a substantial improvement in fit or merely results in overfitting, the procedure shown in Figure 3 was applied.

The one- and two-parameter distributions of the exponential (Exp) and 2-lognormal (LN2) distributions were designated to serve as a reference for evaluating the overparameterization and overfitting in the three- and four-parameter distributions GEV and GGEV.

Figure 3. Workflow for evaluating overparameterization and overfitting in multi-parameter probability distributions.

The first analysis focused on examining whether the theoretical GEV and GGEV distributions significantly alter the shape parameter compared to the LN2 distribution (Fig. 3: Step 2) (Raynal-Villasenor and Raynal-Gutierrez, 2014). This investigation aimed to determine whether the GEV and GGEV distributions are unnecessarily complex (overparameterized), and whether fitting these distributions enhances the accuracy of predictions for extreme values, especially at very long return periods.

In the second analysis, the Kolmogorov-Smirnov test (KS test) was used (Kim et al., 2017) in two testing variants: 1. theoretical quantiles with empirical data and 2. empirical data with random quantiles (Fig.3: Step 3). This was done for the GEV, GGEV, Exp, and LN2 distributions. The hypothesis was that a p-value of less than 0.05 would suggest rejecting the hypothesis that the samples come from the same distribution. The KS test was used to determine whether the multi-parameter distributions (such as GEV and GGEV) might provide slightly better fits in some cases (variant 1) and whether they could be more prone to overfitting (variant 2) (Ozonur et al., 2021). Moreover, sensitivity analysis was also performed by randomly sampling from a two-, three- and four-parameter distribution separately and then testing whether these distributions would not "overturn" and lead to more erroneous extrapolations beyond the range of the empirical data used for calibration. This evaluation was based on quantile-quantile (QQ) plots, in which the quantiles obtained from the calibrated distributions were compared with the empirical ones.

In the third analysis, the K-fold cross-validation (split sample test) was used to validate the distribution's performance (Fig. 3: Step 4) (Kim et al., 2017; Xu and Goodacre, 2018). In this study, we employed the k-fold cross-validation technique, specifically dividing the data series into 5 equal folds (also called 5-folds) (Rohani et al., 2018; Yadav and Shukla, 2016). The distribution is trained on k-1 subsets and tested on the remaining subset. This process is repeated k times until each subset has

been used as the test set (Prusty et al., 2022). K-fold cross-validation is often used for comparing and selecting the best distribution for a given predictive problem. This method allows for evaluating which distribution generalizes best to a new data set (Brunner et al., 2018; Jaiswal et al., 2022). Cross-validation was performed for the GEV and GGEV distributions. To check the results, two measures were used: MAE (for leptokurtic distributions), and RMSE (for platykurtic distributions) (Karunasingha, 2022). As regards the question of how these analyses would be conducted for distributions with fewer than three parameters, two additional distributions – Exp and LN2 – were selected for testing. Finally, a comparison of the cross-validation results between GEV, GGEV and Exp, LN2 was conducted. For the GEV and GGEV, only the distribution with the best fit following the MAE and RMSE, was considered in that analysis. That means that the total number of tested samples was 678 for each of the Exp and LN2 distributions. In contrast, there were 172 samples for the GEV and 281 for the GGEV distribution.

The methods for determining the Exp and LN2 distributions and their goodness-of-fit assessment are presented in the Supplementary Material S1 and S2. The generation of random samples for the Exp, LN2, GEV and GGEV distributions is described in the Supplementary Material S3.

#### 310 4 Results and discussion



### 4.1 Goodness of fit results in relation to the trend category

Among the 678 samples, no trend (*NMT*) was observed in the highest number of cases (446). Conversely, a negative trend (*NT*) was identified in 200 samples, while the least number of samples exhibited a positive trend (*PT*) (32 samples) (Figure 4, Table S1).

Figure. 4. Count of the four fitting distributions (GEV, GGEV, LN3, P3) with marked trend categories.





In the case of NMT, the accuracy measures show: for LN3 – a good fit to 61 samples, for P3 – a good fit to 96 samples, for GEV – a good fit to 111 samples; and for GGEV – a good fit to 178 samples (Figure 4). In the case of NT, the accuracy measures led to fitting the LN3 distribution to 27 samples, the P3 to 32 samples, GEV to 53 samples, and GGEV to 88 samples (Figure 4). In turn, for a PT, the accuracy measures resulted in fitting the P3 distribution to 4 samples, the LN3 – to 5 samples, GEV - to 8 samples, and GGEV - to 15 samples (Figure 4). In NMT samples, the GGEV distribution was the most frequently identified and the LN3 distribution was the least common. Similarly, for the NT and PT samples, the GGEV distribution was most frequently observed. In contrast, the LN3 distribution was the least common in NT samples, while the P3 distribution was the least frequently observed in PT samples (Figure 4). Among the four examined distributions, the GGEV distribution predominates in terms of count for all trend categories (Figure 4, Table S1). This is consistent with the findings by (Nascimento et al., 2015), who established, for the maximum monthly flow data, that the best model was the GGEV rather than the GEV. Focusing solely on the three-parameter distributions (P3, LN3, and GEV), it is evident that the GEV distribution is most frequently fitted best, followed by P3 and LN3. This applies to both NMT and NT samples. In contrast, for the PT samples, the GEV distribution has the highest number of best-fit samples among the three-parameter distributions, followed by LN3 and, finally, P3 (Figure 4, Table S1). This is consistent with Kumar et al. (2003), who argue that, in terms of the L-moments, GEV provided a better fit than P3. Bezak et al. (2014) obtained a completely different result, namely that, in terms of MLE, the best results were obtained with the P3 distribution.

### 4.2. Goodness of fit results in relation to the trend and catchment size categories

Next, it was checked whether a similar pattern of results is obtained when considering the catchment area size ranges (Figure

335 5).

Count of fitting distribution





10-100

œ.

C. 100-1000

1000-10000

E. >10000

A. 0-10

10-100

œ.

A. 0-10

0-10

100-1000

B. 10-100

D. 1000-10000

E. >10000

Figure 5. Count of the four fitting distributions (P3, LN3, GEV and GGEV) with marked trend and catchment size categories.

100-1000

Ö

Catchment area size [km2]

1000-10000

ä

E. >10000

For the *NMT* samples, the best-fitted distribution is the GEV – for samples where the catchment area is less than 10 km<sup>2</sup> or is in the range 10-100 km<sup>2</sup>, or P3 – for those above 10,000 km<sup>2</sup>. Meanwhile, the GGEV distribution is the best fit for samples with catchment areas in the range of 100-1000 km<sup>2</sup> and 1000-10,000 km<sup>2</sup> (Figure 5, table S2). Comparing these results for three-parameter distributions utilizing the MLE estimation method, Gruss et al. (2022) obtained findings for six data series with no trend. Among these, the Weibull, GEV, LN3, and P3 distributions were best fitted to the empirical data from subcatchments with areas ranging from 100 to 1000 km<sup>2</sup>. Moreover, as reported by Gruss et al. (2022), the GEV distribution was fitted best for two catchment areas ranging from 1000 to 10000 km<sup>2</sup>. In this study, in the context of *NMT* samples, the least fitted distributions to the empirical data were: LN3 (samples with catchment areas in the range of 10-100 km<sup>2</sup>, 100-1000 km<sup>2</sup>, and 1000-10,000 km<sup>2</sup>) and additionally LN3 and GGEV for areas larger than 10,000 km<sup>2</sup>.

There are no *NT* samples for catchments smaller than 10 km<sup>2</sup>. The GEV distribution best fits to empirical data from catchments with areas in the range of 10-100 km<sup>2</sup>. Conversely, the GGEV distribution has the best fit for catchments in the range of 100-1000 km<sup>2</sup>, 1000-10,000 km<sup>2</sup>, and above 10,000 km<sup>2</sup> (Figure 5, table S2). This is consistent with Silva and Nascimento (2022) for catchments with areas greater than 10,000 km<sup>2</sup> like the Gurguéia River catchment in Brazil. As reported by the authors, the GGEV distribution has a better fit than the GEV. Gruss et al. (2022) concluded, for Czech Republic and Poland, that the Weibull distribution fits best for catchment areas ranging from 100 to 1000 km<sup>2</sup>, the Weibull and P3 distributions for catchment areas from 1000 to 10,000 km<sup>2</sup>, and the GEV distribution for catchment areas above 10,000 km<sup>2</sup>. In this study, in the context of *NT* samples, the least fitted distributions to the empirical data are P3 (samples with catchment areas in the range of 10-100 km<sup>2</sup>), P3 and LN3 (samples with catchment areas in the range of 100-1000 km<sup>2</sup>), LN3 (samples with catchment areas in the range of 1000-10,000 km<sup>2</sup>), and additionally GEV for >10,000 km<sup>2</sup> (Figure 5, table S2).

The *PT* samples are the scarcest, occurring only in catchments within the range of 10-100 km<sup>2</sup>, 100-1000 km<sup>2</sup>, and 1000-10,000 km<sup>2</sup>. The GGEV distribution fits best for these samples (Figure 5, table S2). In the context of *PT* samples, the least fitted distributions to the empirical data are P3 (samples with catchment areas in the range of 10-100 km<sup>2</sup>), and P3 (samples

with catchment areas in the range of 100-1000 km<sup>2</sup>) and additionally GEV (samples with catchment areas in the range of 1000-10,000 km<sup>2</sup>) (Figure 5, table S2).

## 4.3. Goodness of fit results in relation to the catchment size and peak flow





Next, it was checked whether the relationship between the catchment area (A) and the registered maximum peak flow  $(Q_p)$  could influence the choice of distribution.

Figure 6. Relationship between the catchment area (A) and the peak flow magnitude (Qp) for observational series with no trend and for the four fitting probability distributions (P3, LN3, GEV) and GGEV.

The probability distributions determined for the *NMT* samples show a relationship between *A* and *Qp*, represented by a simple regression line (Figure 6, Table S3). The widest range of *A* is characterized by the samples for which the P3 distribution (30-20,000 km²) and the GEV distribution (3.5-170,000 km²) provide the best fits, while LN3 (35-110,000 km²) and GGEV (50-70,000 km²) fit more limited ranges. Moreover, the widest range of *Qp* is characterized by the samples for which the P3 (1.9-7,000 m³/s) and GEV (1.6-7,000 m³/s) distributions fit best, suggesting that these distributions are the most flexible in modeling extreme flows for different basin sizes. In contrast, the LN3 (8.5 to 6.500 m³/s) and GGEV (2 to 6,000 m³/s) distributions show a more limited applicability (Figure 6, Table S3). Since this may appear to contradict the results presented in the previous sections, a similar analysis is carried out below, but separately for each trend category.

Figure 7. Relationship between the catchment area (A) and the peak flow magnitude (Qp) for observational series with a negative trend and for the four fitting probability distributions (P3, LN3, GEV) and GGEV.

When we focus on the observational series with a NT (Figure 7, Table S4), the widest range of A is characterized by samples fitted to the GGEV distribution (80-180,000 km²), whereas other distributions (P3, LN3, GEV) have more limited ranges. The widest range of Qp is characterized by samples fitted to the GGEV distribution (5-7,000 m³/s), indicating its versatility in modeling extreme flows for samples with a detected NT. The P3 and LN3 distributions have narrower ranges, making them less flexible for samples with a detected NT. This suggests that the GGEV distribution is particularly well-suited for extreme flow events with NT. Moreover, for the NT samples, the GEV distribution fits much better than any of the other three-parameter distributions.




Figure 8. Relationship between the catchment area (A) and the peak flow magnitude (Qp) for observational series with a positive trend and for the four fitting probability distributions (P3, LN3, GEV) and GGEV.

For the PT samples (Figure 8, Table S5), the widest range of area A is characterized by samples fitted to the GEV distribution (48-2,470 km<sup>2</sup>), while other distributions (GGEV, LN3, P3) have more limited ranges. The widest range of Qp is characterized

by samples fitted to the GGEV distribution (8-750 m³/s), indicating its flexibility in modeling extreme flows. The P3 and LN3 distributions have narrower ranges, making them less flexible for samples with a detected PT. In another study on the evaluation of the GEV and LN3 distributions, carried out for the Sewden River, Kousar et al. (2020), using the L-moments estimation, concluded that two locations, one with *A* in the range of 1000-10,000 km², and one with *A* above 10,000 km², exhibit a platykurtic distribution and fit best to the GEV. In turn, the LN3 distribution was the best fit for three other locations that exhibit a leptokurtic distribution for areas ranging 100-1000 km², 1000-10,000 km², and above 10,000 km².

## 4.4 Influence of the environmental factors and sample characteristics on the probability distribution parameters

#### 4.4.1 The GEV distribution





400 Fig. 9. RDA results of the relationship between environmental factors, sample characteristics and the parameters of GEV (σ, ξ, μ), distinguishing between: (a) Kurtosis instead of skewness, (b) Skewness instead of kurtosis, (c) Catchment area ranges instead of variable A and the fourth moment. Descriptions of symbols: μ – location parameter; σ – scale parameter; ξ – shape parameter; catchment area ranges (in km²): a – A < 10; b – 10 ≤ A < 100; c – 100 ≤ A < 1,000; d – 1,000 ≤ A < 10,000; e – A > 10,000; flow peak (Qp); trend (NMT is no trend; PT is positive trend and NT is negative trend); nature of the watercourse (L – lowlands, H – highlands, M – mountains), parameters of the empirical sample (Skew. is empirical skewness, Kurt. is empirical kurtosis and 4thMoment is the 4th center moment); N – sample size.

In the first RDA the first two axes (RDA 1 and RDA 2) explain 80.50% of the variance (63.51% and 16.99%, respectively) (Fig. 9a). In the second RDA, they explain 86.46% (63.60% and 22.86%, Fig. 9b), and in the third - 85.09% (61.75% and 23.34%; Fig. 9c). In the first and second RDA (Fig. 9a-b), the Qp and A are strongly correlated with RDA 1 and Kurt. or Skew. with RDA 2. According to the response variables,  $\sigma$  and  $\mu$  are related to RDA 1, and  $\xi$  is related to RDA 2 (Fig. 9a-b). This is consistent with the findings of Tabari et al. (2021b) and Villarini and Smith (2010). The second and third RDA show that  $\sigma$ and  $\mu$  are inversely proportional to PT (Fig. 9b) and to c (Fig. 9c), respectively. The second RDA shows that  $\sigma$  and  $\mu$  are correlated with N (Fig. 9b). In the third RDA,  $\sigma$  is strongly positively correlated with Qp, which is in line with the findings of Villarini and Smith (2010). This relationship is further supported by the use of scale-invariant statistics, which show good correlations with historical flood-frequency records (Turcotte 1993). This is consistent with Tabari et al. (2021b), who report that the  $\sigma$  parameter of the GEV distribution represents the deviation around the mean and serves as an indicator of variance. However, it is important to note that  $\sigma$  can vary over time, as demonstrated by the application of a non-stationary GEV model to account for changing streamflow series (Jiang and Kang, 2019). Moreover, in the third RDA,  $\mu$  is strongly positively correlated with e and 4thMoment (Fig. 9c). The  $\mu$  parameter indicates the center of the distribution, acting as an indicator of the mean. What is particularly noteworthy is that hydrological signatures related to flow magnitude, such as  $\mu$  and  $\sigma$ , are primarily dependent on A (Fig. 9a-b), which significantly influences their values, while other attributes have a lesser impact on the response variable. This is consistent with previous findings by He et al. (2015), Northrop (2004), and Tyralis et al. (2019). Another study using the MLE method confirms the above, indicating a linear relationship between  $\mu$  and  $\sigma$ , which means that as the catchment area increases, so do these parameters (Northrop, 2004). Current research confirms this trend (Fig.

9a-b). In sample e, which is a very large catchment, 4thMoment affects the  $\mu$  parameter. Sample c with PT is weakly negatively 425 correlated with both  $\mu$  and 4thMoment. Other catchment types (a, b, and d) have a weak influence on the parameters of this distribution.  $\xi$  is positively related to Kurt., H, M, and NMT, Skew. (Fig. 9a-b) and a (Fig. 9c). The  $\xi$  parameter determines the tail behavior of the distribution (He et al., 2015). Specifically, higher values of  $\xi$  lead to heavier tails (Tabari et al., 2021b; Tyralis et al., 2019; Villarini and Smith, 2010). The RDA reveals that  $\zeta$  is inversely proportional to L and NT (Fig. 9a-b) and 430 additionally to d (Fig. 9c). This is consistent with other research (Ahilan et al., 2012; He et al., 2015; Sampaio and Costa, 2021; Tabari et al., 2021b; Tyralis et al., 2019; Villarini and Smith, 2010). However, other researchers did not indicate that the  $\xi$ parameter could be related to the type of landscape. Morrison and Smith (2002) found that  $\xi$  is not dependent on the basin morphological parameters or land cover properties, suggesting that other factors may be at play. Kumar et. al (2003) highlight the importance of the GEV distribution in regional FFA, but do not specifically address the relationship between  $\xi$  and the highlands area. Nonetheless,  $\zeta$  is more likely linked to hydrological processes and meteorological conditions than to A (He et 435 al., 2015). However, other studies have indicated some correlations between  $\xi$  and either terrain elevation or the type of landscape. The  $\xi$  parameter of the GEV distribution is correlated with nature (terrain elevation) (Sampaio and Costa, 2021; Tyralis et al., 2019). Moreover, research confirms that the type of landscape affects the distribution of GEV (Ahilan et al., 2012). The magnitude of  $\xi$  in the GEV distribution depends on  $\mu$  of the gauge, irrespective of it being in lowlands, highlands, 440 or mountains (Villarini and Smith, 2010). According to Tyralis et al. (2019),  $\xi$  exhibits a negative linear correlation with the catchment mean elevation. Which means that as the elevation increases, the value of  $\xi$  slightly decreases. Our study actually indicates an opposite trend (Fig. 9a-c). However, the impact of the morphologic characteristics of the catchments in the regression model for the GEV  $\xi$  parameter is small (Sampaio and Costa, 2021). This is also confirmed by the current study landscape forms have a weak influence on the parameters of this distribution. The  $\zeta$  parameter dependency is mainly influenced 445 by climatic indices, while other catchment characteristics are less significant (Tyralis et al., 2019). This is consistent with He et al. (2015) who found no relationship between  $\xi$  and A, suggesting that the hydrological heterogeneity is implicitly captured by  $\xi$ . In the first RDA N is correlated with RDA 3, in the third RDA N and b are related to RDA 3 (Fig. 10c).

## 4.4.2 The GGEV distribution





In the first RDA the first two axes (RDA 1 and RDA 2) explain 54.00% of the variance (45.76% and 8.24%, respectively) (Fig. 10a). In the second RDA, they explain 54.36% (45.78% and 8.58%; Fig. 10b), and in the third – 56.19% (47.19% and 9.00%; Fig. 10c). Similar to the GEV distribution, in both Fig. 10a and 10b,  $\sigma$  and  $\mu$  (which are related to the RDA1 axis) are strongly positively correlated with Qp and A. In contrast, in the third RDA (Fig. 10c),  $\sigma$  remains strongly positively correlated with Qp, while  $\mu$  is strongly positively correlated with e and 4thMoment. This means that higher Qp values correspond to a larger  $\sigma$  parameter in the GGEV distribution. Larger catchment areas e lead to an increase in  $\mu$ , which shifts the central point of the distribution. Across all redundancy analyses (Fig. 10a–c),  $\xi$  and  $\delta$  show a positive relationship with *Kurt*. or *Skew*., which may support the additional parameter mechanism described by Nascimento et al. (2015) for this distribution. Moreover,  $\xi$  and  $\delta$  show a positive relationship with NMT (Fig. 10a-c). Observing the biplots (Fig. 10a, 10c), it is noted that the Kurt. parameter affects  $\xi$  and  $\delta$ , while 4thMoment influences  $\mu$ . An increase in 4thMoment, which measures the concentration of values around the mean and is also related to Kurt., indicates an increase in  $\mu$ . The  $\mu$  parameter determines where the center of the distribution is located on the number line. The greater 4thMoment, the higher  $\mu$  in a heavy-tailed distribution. This means that, where more extreme values occur, the central tendency of the distribution (measured by the  $\mu$  parameter) shifts towards these higher values to better reflect the influence of extremes on the distribution (Tabari et al., 2021b). Since 4thMoment is not associated with RDA2, it will not directly influence  $\xi$  and  $\delta$ , or its impact will be limited for the samples examined (Fig. 10a). However, 4thMoment used to determine Kurt. will cause Kurt. to strongly correlate with  $\xi$  (Fig. 10c).  $\xi$  is correlated with Skew. and Kurt. of the empirical data. This means that  $\xi$  influences the asymmetry and tail distribution of empirical flow data, which is consistent with the description by (Nascimento et al., 2015). In this study  $\xi$  and  $\delta$  are inversely related to N. In practice, this might suggest that with larger N, the distribution becomes less extreme or lighter. The shape parameters likely adjust to reflect a more stable and less variable distribution as the amount of data increases. The third RDA additionally revealed that  $\xi$  and  $\delta$  are negatively correlated with NT (Fig. 10c). This may indicate that in situations where there is a downward trend in the data, the distribution becomes less varied or more flattened. A weak correlation could suggest that with a NT, the values of  $\delta$  and  $\xi$  may slightly decrease. We observe that  $\delta$  is not as strongly correlated with Skew. as  $\xi$ .  $\delta$  serves as a supplementary parameter, and the canonical analysis shown in Fig. 10a-b indicated that  $\delta$  has similar properties to  $\xi$ . Meanwhile, 3thMoment strongly correlates with  $\sigma$  (not shown on the graph) because  $\sigma$  affects the magnitude of deviations from the mean, and the 3rd central moment measures precisely these deviations.  $\sigma$  and  $\mu$  are negatively correlated with c (Fig. 10c). The other independent variables (H, L, M, PT, b, d) are associated with RDA3 and do not affect the distribution parameters. This suggests that terrain topography does not have a direct impact on the parameters of GGEV. Additionally, it can be suggested that not all the types of catchments influence the shaping of distribution parameters. Very large catchments (e) have a strong positive impact, while c have only a weak influence, and there is no effect on the parameters of d and d. This may be because distribution parameters (in this case  $\mu$ ) affecting larger areas may not have as strong an impact on smaller catchments, where local effects dominate over the effects associated with distribution parameters (Arnaud et al., 2011; Roodsari and Chandler, 2017).

Fig. 10. RDA results of the relationship between environmental factors, sample characteristics and the parameters of GGEV  $(\sigma, \xi, \delta, \mu)$ , distinguishing between: (a) Kurtosis instead of skewness, (b) Skewness instead of kurtosis, (c) Catchment area ranges instead of variable A and the fourth moment. Descriptions of symbols:  $\mu$  – location parameter,  $\sigma$  – scale parameter,  $\xi$  – shape parameter,  $\delta$  – shape parameter of GGEV extension, catchment area ranges (in km²): a - A 

Fig.11. RDA results of the relationship between environmental factors, sample characteristics and the parameters of P3  $(\sigma, \xi, \mu)$ , distinguishing between: (a) Kurtosis instead of skewness, (b) Skewness instead of kurtosis, (c) Catchment area ranges instead of variable A and the fourth moment. Descriptions of symbols:  $\mu$  – location parameter,  $\sigma$  – scale parameter,  $\xi$  – shape parameter, catchment area ranges (in km²): a - A < 10;  $b - 10 \le A < 100$ ;  $c - 100 \le A < 1,000$ ;  $d - 1,000 \le A < 10,000$ ; e - A > 10,000; flow peak  $(Q_p)$ ; trend (NMT) is no trend; PT is positive trend and PT is negative trend); nature of the watercourse (L - lowlands, H - highlands, M - mountains), parameters of the empirical sample (Skew) is empirical skewness, E0 sample size.

The first two axes (RDA 1 and RDA 2) explain 66.72% of variance (62.38% and 4.34%, respectively) (Fig. 11a). In the second RDA, they explain 66.72% (62.35% and 10.40%, respectively) (Fig. 11b), and in the third – 73.30% (61.78% and 11.52%, respectively) (Fig. 11c). A comparison of the RDA results for the 3P distribution parameters with those of GEV and GGEV has revealed the following dependencies:  $\xi$  is strongly negatively correlated with N and weakly with Kurt. and Skew. (Fig. 11a-c). This means that with smaller sample sizes and higher Skew., the  $\xi$  parameter is larger. This is confirmed by the study of Hu et al. (2020). As reported by Hu et al. (2020), Skew. in the log-P3 distribution is very sensitive to N. In turn,  $\xi$  is weakly positively correlated with L and negatively correlated with L (Fig. 11a-b). The L (Fig. 11b). L is weakly correlated with L and L (Fig. 11b). In the second RDA, L is inversely proportional to L (Fig. 11c). The L parameter has a weak positive relation to the L parameter (Fig. 11c). In catchments L increases. In contrast, L (L (L (L )), L (L ) and L (L ) and L (L ). In catchments L (L ) increases. In contrast, L (L ), L (L ), L (L ), L (L ) and L (L ) and L (L ).

#### 4.4.4 The LN3 distribution







Fig. 12. RDA results of the relationship between environmental factors, sample characteristics and the parameters of LN3  $(\sigma, \xi, \mu)$ , distinguishing between: (a) Kurtosis instead of skewness, (b) Skewness instead of kurtosis, (c) Catchment area ranges instead of variable A and the fourth moment. Descriptions of symbols:  $\mu$  – location parameter,  $\sigma$  – scale parameter,  $\xi$  – shape parameter, catchment area ranges (in km²): a - A < 10;  $b - 10 \le A < 100$ ;  $c - 100 \le A < 1,000$ ;  $d - 1,000 \le A < 10,000$ ; e - A > 10,000; flow peak  $(Q_p)$ ; trend (NMT) is no trend; PT is positive trend and PT is negative trend); nature of the watercourse (L - lowlands, H - highlands, M - mountains), parameters of the empirical sample (Skew) is empirical skewness, E0 sample size.

The first two axes (RDA 1 and RDA 2) explain 46.61% of the variance (38.71% and 7.90%, respectively) (Fig. 12a). In the second RDA, they explain 54.09% (38.98% and 15.11%, respectively) (Fig. 12b), and in the third – 57.72% (42.60% and 15.12%, respectively) (Fig. 12c). A comparison of the RDA results for the LN3 distribution parameters with those of GEV, P3, and GGEV has revealed the following dependencies:  $\mu$  is weakly correlated with *NMT* (Fig. 12a-b). *PT*, *NT*, *M*, and *L* are correlated with RDA 3 (Fig. 12a-b). In the first RDA,  $\xi$  is negatively related to *Kurt*. and *H* is weakly correlated with *N*. (Fig. 12a). In the second RDA,  $\xi$  is negatively related to *Skew*. and *H* (Fig. 12b-c).  $\mu$  is weakly inversely proportional to *N* (Fig. 12b). In the third RDA, in contrast to GEV, GGEV, and P3,  $\sigma$  is strongly positively correlated with e and e is strongly positively correlated with e and e is negatively correlated with *NMT*. Moreover, e is negatively correlated with *N* and *NT* (Fig. 12c). e is negatively correlated with e and e is negatively correlated with e is negatively e in e in

### 4.4.5 Key points on the influence of environmental factors and sample characteristics

The following points summarize key findings regarding the relationships between environmental factors and sample characteristics and the parameters of the studied probability distributions:

- GGEV tends to have the H, M, L located outside the RDA1 and RDA2.
- GGEV and P3 share a common feature of a negative correlation between N and  $\xi$ , while GEV and LN3 exhibit more complex correlation patterns.
- GGEV, GEV, and P3 show similar correlations between the 4thMoment and Qp, as well as e in the context of RDA1, whereas LN3 differs in this respect.
  - Both GGEV and GEV exhibit a pattern characterized by a negative correlation between NT and  $\xi$ , and a positive correlation between  $\xi$  and NMT.
  - Only for the P3 distribution is the  $\xi$  parameter more strongly negatively correlated with N than with Kurt. and Skew. This confirms that the  $\xi$  parameter is highly dependent on N.
- Only the patterns for GGEV and LN3 show a *PT* along the RDA3.
  - Catchment size types influence the distribution parameters, with the most types affecting the parameters of GEV and the least types affecting the parameters of GGEV, LN3 and P3.

Based on the above comparison, the GGEV distribution shows some similarities with other distributions regarding the occurrence and correlation of the distribution parameters. However, there are differences in certain aspects, such as the distribution of parameters in the principal components and parameter correlations, which indicates unique characteristics of GGEV compared to GEV, LN3, and P3. GGEV often differs from other distributions in how its parameters spread within the principal component space, which may be significant when modeling and interpreting the results of extreme flow data analysis.

## 4.5. Overparameterization check

In order to evaluate the overparameterization or overfitting problem, the results are summarized below according to the 4 steps of the methodology outlined in Figure 3.

## Step 1


Out of the two distributions (Exp and LN2), only LN2 demonstrates the best fit across all the 678 profiles based on accuracy measures and scoring rules. Nevertheless, we performed the tests for both distributions. Additionally, these distributions were evaluated against three- and four-parameter distributions (GEV and GGEV) using the same criteria.

### 555 Step 2


For the GEV distribution, analyzed for 172 profiles, the  $\xi$  parameter values are consistently near zero, well below one. In turn, the fitted LN2 distribution has a scale parameter ( $\sigma$ ) greater than 1 for only seven stations, which indirectly affects the distribution's shape. In contrast, the GGEV distribution has  $\xi$  parameter value greater than 1 for a smaller number of four stations. It is worth noting that the  $\delta$  parameter reached a value greater than 1 for 45 profiles out of the 281 analyzed. However, as shown by the RDA analysis, the contribution of the  $\delta$  parameter relative to the  $\xi$  parameter is smaller. This may suggest that the  $\delta$  parameter primarily compensates for the limitations of the  $\xi$  parameter in distributions with lighter tails. More details can be found in the Supplementary Materials, in Tables S6-S8. This is confirmed by the research of Nascimento et al. (2015), who state that when the  $\delta$  parameter is less than 1, the GGEV exhibits a heavier tail than the GEV, making it more effective for modeling extreme events, which may occur more frequently than lighter-tailed distributions would predict.

Fig. 13. Empirical (black dots) and theoretical probability density functions of GEV and GGEV for *Qp*, shown for three profiles: (a) 149180130, (b) 149180230, (c) 149190010.

In order to investigate how the  $\delta$  parameter in the GGEV distribution affects the form of the extreme value distribution, a visual inspection was conducted (Fig. 13). Figure 13 shows the ability of both GEV and GGEV distributions to model hydrological extremes. At profile 149220070, the GGEV captures the peak of the distribution more accurately than GEV. For station 150190080, the GGEV captures the peak of the distribution more accurately and GGEV provides a slightly better fit in the upper tail. At station 150200090, both models follow the empirical curve well, although GGEV shows a marginal advantage in modeling the higher extremes.

## Step 3







575 The KS test, which compares the theoretical quantiles (Exp distribution) with the empirical data, found no significant differences (p-value > 0.05) for 372 out of 678 profiles, indicating agreement between the distributions. Similarly, when comparing empirical quantiles to random samples (Exp distribution), 279 profiles showed no significant differences, suggesting a comparable nature of the empirical and random distributions in these cases. This corresponds to a fit rate of 74.9% (279/372).

A better fit was obtained for the LN2 distribution. The KS test comparing the theoretical quantiles (LN2 distribution) with the empirical data found no significant differences (p-value > 0.05) for 678 out of 678 profiles, indicating agreement between the distributions. Similarly, when comparing empirical quantiles to random samples (LN2 distribution), 661 profiles showed no significant differences, suggesting a comparable nature of the empirical and random distributions in these cases. This corresponds to a fit rate of 97.5% (678/661).

A much better fit was obtained for the GEV and GGEV distributions. The KS test comparing the theoretical quantiles (GEV distribution) with the empirical data found no significant differences (p-value > 0.05) for 172 out of 172 profiles, indicating agreement between the distributions. Similarly, when comparing empirical quantiles to random samples (GEV distribution), 171 profiles showed no significant differences, suggesting a comparable nature of the empirical and random distributions in these cases. This corresponds to a fit rate of 99.4% (171/172). The KS test comparing theoretical quantiles (GGEV distribution) with empirical data found no significant differences (p-value > 0.05) for 281 out of 281 profiles, indicating agreement between the distributions. Similarly, when comparing empirical quantiles to random samples (GGEV distribution), 281 profiles showed no significant differences, suggesting a comparable nature of the empirical and random distributions in these cases. This corresponds to a fit rate of 100% (281/281).

The results showed that distributions with more parameters (three or more, such as GEV and GGEV) not only provided a slightly better fit in some cases (in the empirical data vs. theoretical quantiles scenario) but were also less prone to overfitting (in the empirical data vs. random quantiles scenario) This is confirmed in the QQ plots; see the Supplementary Material.

#### Step4

The GEV and GGEV distributions were subjected to the K-Fold cross-validation alongside the LN2 and Exp distributions. The K-Fold cross-validation results are presented as the percentage distribution of outcomes across individual intervals relative to the total number of results (Table 1).

Table 1. Percentage distribution of the K-Fold cross-validation results across individual intervals relative to the total number of outcomes.

| outcomes     |         |            |         |                 |      |          |
|--------------|---------|------------|---------|-----------------|------|----------|
|              | MAE     | MAE        | RMSE    |                 | RMSE | (1001-10 |
| Distribution | (0-100) | (101-1000) | (0-100) | RMSE (101-1000) | 000) |          |
| 2LN          | 94.8%   | 5.22%      | 92.37%  | 7.63%           |      | 0%       |
| Exp          | 94.8%   | 5.22%      | 92.37%  | 4.24%           |      | 3.39%    |
| GEV          | 95.09%  | 4.91%      | 100%    | 0%              |      | 0%       |
| GGEV         | 94.9%   | 5.12%      | 98.48%  | 1.51%           |      | 0%       |

Explanation: MAE – mean absolute error, RMSE - root mean square error, 0-100 – best-fitting model, 101-1000 – well-fitting model, 1001-10,000 – poorest-fitting model.

MAE and RMSE values vary significantly, ranging from very low (close to 0) to much higher values (e.g., 1000). High values suggest that the model predicts river flows less accurately for certain rivers. The intervals represent the quality of model fit, with 0-100 indicating the best fit, 101-1000 well fitting model, and 1001-10,000 the poorest fit. The GEV distribution achieved the highest percentage of best-fitting models (95.09% for MAE and 100% for RMSE), indicating superior performance compared to the other distributions. The GGEV distribution also showed strong results, with 94.9% of models falling in the best-fit category for MAE and 98.48% for RMSE. The LN2 and Exp distributions performed similarly, with over 94% of results in the best-fit category for both MAE and RMSE. However, the Exp distribution showed a small proportion (3.39%) of poorest-fitting models in the RMSE category, which was not observed for the other distributions. To summarize, Step 4 has shown that the GGEV and GEV distributions have excellent predictive efficiency (better than that of the distributions with fewer parameters), which demonstrates that in most cases analyzed, they are quite robust to overparameterization and overfitting.

Although the study used observational series of 30 years or more, the number of profiles analyzed in highland and mountainous areas was considerably lower than that in lowland areas. Furthermore, the number of observational series exhibiting a positive trend in the analyzed region was limited; moreover, the study did not account for the non-stationarity in the parameters of the analyzed distributions.

## **5 Conclusions**







The main findings of this research can be summarized as follows:

- 1. Based only on the trend criterion, the GGEV distribution, compared to the analyzed three-parameter distributions, and the GEV distribution compared to the other three-parameter distributions were the best fit for most samples.
- 2. Based on the trend criterion and the catchment size, it was found that the GEV distribution is best suited for microand meso-catchments, while the GGEV distribution is ideal for macro- to large-catchments, where the series exhibits
  a trend (either negative trend or no trend). The P3 distribution is preferred for very large catchments, but only when
  the sample has no trend. In contrast, for samples with a positive trend, the GGEV distribution performs best across
  the meso- to very large catchments.
  - 3. Compared to the analyzed P3, LN3, and GEV distributions, the GGEV distribution was not flexible regarding both the catchment area and the peak flow for samples with no trend and positive trend, but it was flexible for samples, in which a negative trend was detected.
  - 4. Our findings revealed certain patterns in the shape parameter between the P3 and GGEV distributions, as well as between the GEV and GGEV distributions. Additionally, patterns were noted in the μ parameter among the P3, GEV, and GGEV distributions. Our study also showed that GGEV was the only distribution for which the parameters were not correlated with the forms of landscape (lowlands, highlands, mountains).

- 5. It has been confirmed that adding the shape parameter of the GGEV distribution primarily compensates for the limitations of the shape parameter in distributions with lighter tails.
- 6. Using the Kolmogorov-Smirnov test, it was found that GEV and GGEV not only provided a slightly better fit in some cases (in the empirical data vs. theoretical quantiles scenario) but were also less prone to overfitting (in the empirical data vs. random quantiles scenario and in the theoretical quantiles vs. random quantiles scenario) in comparison to Exp and LN2. Furthermore, the robustness of GEV and GGEV distributions to overparameterization and overfitting was confirmed by the K-Fold cross validation.
- 7. Based on the above, in the era of climate change, distributions such as GGEV are expected to be better suited when trends are present, offering a clear performance advantage.

The results of this study highlight several promising avenues for future research. One potential direction is the further exploration of the GGEV distribution in the context of various hydrological and meteorological phenomena. Given its superior performance in fitting most samples and its sensitivity to trends, especially under non-stationary conditions such as climate change, future studies could examine its applicability across different geographical regions and climatic conditions.

The findings on the influence of catchment types on distribution parameters indicate that more research is needed to refine our understanding of how landscape characteristics interact with hydrological distributions. A deeper exploration into the relationship between the catchment area characteristics, especially in varied topographies and land-use patterns, could yield more universal insights. Expanding the range of predictor variables used in modeling, beyond trend detection, nature of catchment, catchment area, and the hydrological characteristics, might also improve the accuracy and flexibility of distribution selection.

As part of future research directions, the team plans to focus on modeling the GGEV distribution in a full Bayesian approach, involving the estimation of the four parameters of this distribution using MCMC sampling, instead of the ggevp function. In particular, it is planned to determine prior distributions for each of the parameters and to implement the MCMC algorithms, taking into account the need to adjust the proposal distributions, the length of the burn-in phase, thinning, and the number of samples — depending on the specifics of the model under study.

# **Author contributions**





LG: Conceptualization, Data curation, Formal analysis, Investigation, Methodology, Resources, Software, Writing – original draft preparation, Writing – review & editing; PW: Conceptualization, Methodology, Supervision, Validation, Writing – review & editing; PT: Investigation, Resources, Visualization, Writing – original draft preparation, Writing – review & editing; CM: Data curation, Writing – review & editing; MW: Supervision, Validation; JPJr: Conceptualization, Supervision; JPSen: Supervision; Scz: Supervision, Validation.

## **Competing interests**

The authors declare no conflict of interest.

#### Acknowledgments

The authors would like to express their sincere gratitude to Karol Misdziol, MSc Eng. for his help with coding in R and to the Institute of Meteorology and Water Management—National Research Institute in Warsaw for their release of the flow data.

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
