# Peer review of "Evaluation of the Dual Gamma Generalized Extreme Value Distribution for flood events in Poland"

_EGUsphere, 2025_

## Referee Comment (RC2)

This manuscript evaluates the applicability of the Dual Gamma Generalized Extreme Value (GGEV) distribution for flood frequency analysis using data from a large set of catchments in Poland. While the study addresses a relevant topic and applies a technically interesting distribution, major revisions are needed before publication—particularly regarding the inconsistent use of estimation methods (Bayesian vs. MLE), limited justification for the choice of GGEV, excessive detail in the redundancy analysis, and recurring issues with redundancy, figure quality, and sentence clarity, among others. Detailed comments are provided below.

**Mayor comments**

1) The current title, "The application of new distribution in determining extreme hydrologic events such as floods", is somewhat vague and potentially misleading, as it implies that the distribution was developed or derived as part of this study. However, the manuscript focuses on applying an existing distribution—the Dual Gamma Generalized Extreme Value Distribution (GGEV)—to flood frequency analysis.
   I suggest revising the title to name the distribution, improving clarity and precision explicitly. For example, "Application of the Dual Gamma Generalized Extreme Value Distribution to Extreme Hydrologic Events" would more accurately reflect the scope and contribution of the paper.

2) The manuscript adopts the GGEV distribution as a flexible alternative to traditional three-parameter models for flood frequency analysis under non-stationary conditions. However, the rationale for selecting this distribution and the hydrologic interpretation of its parameters—particularly the additional shape parameter (delta)—could be more clearly articulated.
   It would strengthen the manuscript if the authors provided a more detailed explanation of how the GGEV distribution improves the modeling of extreme events, for example, by showing how delta influences tail behavior or skewness in a hydrologically meaningful way. Comparative visualizations (e.g., PDFs or return level plots) could also help clarify the added value of this distribution in the context of practical flood risk applications.

3) The study employs Bayesian MCMC estimation for the GGEV parameters, while using maximum likelihood estimation (MLE) for the GEV, LN3, and P3 distributions. This inconsistency introduces potential biases in model comparison and undermines the objectivity of the goodness-of-fit evaluation.
   It is strongly recommended that the authors either apply a uniform estimation framework across all distributions (e.g., MLE or Bayesian) or explicitly justify using Bayesian inference for the GGEV distribution alone. If the Bayesian approach is retained for GGEV, it would be important to include appropriate posterior diagnostics (e.g., trace plots, convergence statistics) and sensitivity analyses regarding prior assumptions.
   Additionally, in the context of Bayesian modeling, it may be more appropriate to incorporate probabilistic performance metrics that reflect both accuracy and reliability —such as the Continuous Ranked Probability Score (CRPS) or its skill score version (CRPSS) (Hersbach, 2000; Ossandón et al., 2022)— rather than relying solely on point-estimate-based metrics like MAE or RMSE. This would provide a more robust basis for comparing predictive performance across models.

4) While the authors apply the Mann-Kendall test to identify monotonic trends in the data, the modeling framework does not appear to explicitly incorporate non-stationarity in the distribution parameters. This methodological gap could limit the study's relevance in the context of climate change, where the assumption of stationarity is increasingly untenable.
   If the authors choose not to implement non-stationary models (e.g., distributions with time-varying parameters such as mu(t) or sigma(t)), it would be important to include a discussion justifying this decision. Clarifying whether the approach is intentionally based on stratified modeling (e.g., trendbased grouping) rather than a fully process-based framework would help readers understand the scope and limitations of the results, particularly in terms of their applicability to future climate-influenced extremes.

5) While the RDA section is methodologically comprehensive, it currently presents an excessive level of detail that overwhelms the reader and hinders the extraction of key insights. The narrative includes lengthy descriptions of biplots and correlation patterns for each distribution, much of which could be streamlined.

   I suggest briefly introducing how to interpret RDA results in the Methods section (3.5) and then making the Results section (4) more concise and targeted, focusing only on the most relevant and interpretable patterns. Analyzing each RDA result in detail within the main text is unnecessary. For example, the detailed plots and discussion related to the LN3 and P3 distributions could be moved to the Supplementary Material, especially since the primary focus of this study is the application of the GGEV distribution in flood frequency analysis.

   This would allow the main text to highlight the most critical findings while avoiding redundancy and improving readability.

6) The manuscript has recurring issues related to redundancy and incomplete or unclear sentence structure. In several instances, information is repeated across consecutive lines with minimal added value, affecting the text's overall conciseness and flow. Additionally, some sentences appear grammatically incomplete or are phrased in a way that makes their interpretation ambiguous. These issues may hinder reader comprehension and should be systematically addressed.

   For specific illustrations of these concerns, please refer to the minor comments. A thorough language and structural revision is recommended to enhance the manuscript's clarity and readability.

7) The overall quality and resolution of the figures should be improved to enhance their visual clarity and readability. For instance, in Figure 10, it is difficult to clearly discern the horizontal and vertical lines corresponding to zero on each axis. The axis tick labels, as well as any other textual elements within the figure, appear blurry and are challenging to read. This issue applies not only to Figure 10 but also affects several other figures throughout the manuscript. Enhancing the resolution and font clarity would significantly improve the interpretability of the graphical results and ensure they meet publication standards.

**Specific comments:**

1) L48: Lognormal is a two-parameter distribution, or do you mean LN3?. Please correct it if it corresponds.
2) L82: A linear trend means nonstationarity of the time series. Please check it.
3) L98: Please correct the citation. It should be (Silva and Do Nascimento, 2022).
4) L124-125: Please consider improving the phrasing in those lines for clarity.
5) L144: You already mentioned it in L140. Please avoid repetition.
6) L226: Canoco 5.12 is an R package? Please specify.
7) Figure 3. It should say "value of the shape." In the same figure, it should be "generating random samples." Please correct them.
8) L258: Include the reference for Brunner et al. (2018) in the reference list. Please check it for the rest of the references.
9) L287: Generalized is redundant in "the generalized GGEV". Please remove it.
10) L291: There is an extra comma; please remove it.
11) Figure 6. In the legend, it should be "P3" instead of "3P3"
12) Lines 324–325 appear to repeat information already stated in the preceding paragraph. Consider revising this section to avoid redundancy and improve the flow of the text.

13) L336-338: Please revise the phrasing in the sentence. The sentence appears incomplete or abruptly truncated, and the comparison between the distributions could be clarified for better readability.
14) Lines 338–340 appear to repeat information already stated in the preceding lines. Consider revising this section to avoid redundancy and improve the flow of the text.
15) L355-356: it is stated that the widest range of catchment area (A) is represented by samples fitted to the GGEV distribution. However, based on Figure 8, the GEV distribution spans the widest range of A (~50 - ~2,500 km²). Please revise the description to reflect this more accurately.
16) L359: the same as 15). Redundant.
17) L370: Define the term "add. Shape."
18) L423: Add an space before (Fig9. c)
19) L456-457: The sentence "It exhibits a strong correlation with the scale and location parameters, whereas this relationship is not observed for the shape parameter (Tabari et al., 2021b)" is unclear. It is unclear what "It" refers to or which variable is being described as correlated. Please clarify the sentence and ensure that the subject and its relationship to the distribution parameters are explicitly stated.
20) Figures 9 and 10: It looks like for GEV, the shape parameter is less sensitive to the sample size (N) that the shape and add. shape for GGEV. If this is the case, it should be highlighted.
21) L531: repetition. You already said it in L532.
22) L627: In the sentence "the fitted LN2 distribution has a shape parameter value greater than 1 for only seven stations", there appears to be a conceptual error. As far as I understand, the two-parameter log-normal distribution (LN2) includes only a location (mu) and a scale (sigma) parameter and does not have an explicit shape parameter. Please clarify whether this refers to sigma being interpreted as a shape proxy or if there has been confusion with another distribution.

**References**

Hersbach, H. (2000). Decomposition of the continuous ranked probability score for ensemble prediction systems. *Weather and Forecasting*, *15*(5), 559–570. https://doi.org/10.1175/1520-0434(2000)015<0559:DOTCRP>2.0.CO;2

Ossandón, Á., Rajagopalan, B., Tiwari, A. D., Thomas, T., & Mishra, V. (2022). A Bayesian Hierarchical Model Combination Framework for Real-Time Daily Ensemble Streamflow Forecasting Across a Rainfed River Basin. *Earth's Future*, *10*(12), e2022EF002958. https://doi.org/10.1029/2022EF002958

---

## Author Response (AR1)

**Dear Referees,**

we would like to express our sincere gratitude for your insightful and constructive review of our manuscript. Your comments were extremely helpful and have significantly improved the quality of our work. We carefully addressed all of the issues you pointed out, and we believe that the revised manuscript now presents a more comprehensive and robust analysis of our research. We appreciate your recognition of the novel features of our work and your positive assessment. We are confident that the revised manuscript will be of interest to the journal's readership. Thank you again for your time and effort. We have revised and improved this manuscript accordingly. Specifically, we have corrected the following sections: (1) Title (2) Abstract (3) Introduction, (4) Materials and methods, (5) Results and Discussion, (6) Conclusions (7) References. We have also revised and improved the Supplementary material. Before resubmission, the entire article was reviewed by a native speaker of American English. All changes to the manuscript have been incorporated and are marked in red. Below, we provide a detailed explanation of how we addressed each of your comments, along with the line numbers where modifications or additions were made.

Kind regards,
Łukasz Gruss
(on behalf of all the co-authors)

**Referee 1**

**General comments:**

The authors used data from 678 water gauges in 340 rivers in temperate climate regions of Poland to compare the performance of several probability distributions—including Log-Normal, Pearson Type III, Generalized Extreme Value (GEV), and Dual Gamma Generalized Extreme Value Distribution (GGEV). Overall, the manuscript addresses several scientific questions; however, the following issues need further clarification:

**Specific comments:**

1. The term "new distribution" in the title is vague and should be clarified.

**Response:** Thank you for pointing this out. The title "The application of new distribution in determining extreme hydrologic events such as floods" has been replaced with "Evaluation of the Dual Gamma Generalized Extreme Value Distribution for flood events in Poland". This will help to clarify the main objectives of our research.

2. Section 4.4 contains excessive text; it is recommended to streamline the analysis and reduce unnecessary descriptions.

**Response:** We thank the Referee for pointing this out. Subsection 4.4 has been thoroughly reviewed and revised. Each subsequent analysis for the respective distributions now includes only the information that differs from the previous one, in order to avoid unnecessary repetition. We hope that the revised version is clearer and more accessible compared to the previous one.

3. The conclusion section includes some repetition and wordiness. For example, point 7 "It was found that adding the shape parameter of the GGEV distribution primarily compensates for the limitations of the shape parameter in distributions with lighter tails" reflects a known characteristic of the GGEV distribution and is not appropriate to be highlighted as a study conclusion.

**Response:** We thank the Referee for pointing this out. The conclusions have been revised. Below, we present the updated version copied from the manuscript. We believe that the revised conclusions now better reflect the findings of our study.

**"5 Conclusions**

The main findings of this research can be summarized as follows:

- 1. Based only on the trend criterion, the GGEV distribution, compared to the analyzed three-parameter distributions, and the GEV distribution compared to the other three-parameter distributions were the best fit for most samples.
- 2. Based on the trend criterion and the catchment size, it was found that the GEV distribution is best suited for micro- and meso-catchments, while the GGEV distribution is ideal for macro- to large-catchments, where the series exhibits a trend (either negative trend or no trend). The P3 distribution is preferred for very large catchments, but only when the sample has no trend. In contrast, for samples with a positive trend, the GGEV distribution performs best across the meso- to very large catchments.
- 3. Compared to the analyzed P3, LN3, and GEV distributions, the GGEV distribution was not flexible regarding both the catchment area and the peak flow for samples with no trend and positive trend, but it was flexible for samples, in which a negative trend was detected.
- 4. Our findings revealed certain patterns in the shape parameter between the P3 and GGEV distributions, as well as between the GEV and GGEV distributions. Additionally, patterns were noted in the μ parameter among the P3, GEV, and GGEV distributions. Our study also showed that GGEV was the only distribution for which the parameters were not correlated with the forms of landscape (lowlands, highlands, mountains).
- 5. It has been confirmed that adding the shape parameter of the GGEV distribution primarily compensates for the limitations of the shape parameter in distributions with lighter tails.
- 6. Using the Kolmogorov-Smirnov test, it was found that GEV and GGEV not only provided a slightly better fit in some cases (in the empirical data vs. theoretical quantiles scenario) but were also less prone to overfitting (in the empirical data vs. random quantiles scenario and in the theoretical quantiles vs. random quantiles scenario) in comparison to Exp and LN2. Furthermore, the robustness of GEV and GGEV distributions to overparameterization and overfitting was confirmed by the K-Fold cross validation.
- 7. Based on the above, in the era of climate change, distributions such as GGEV are expected to be better suited when trends are present, offering a clear performance advantage.

The results of this study highlight several promising avenues for future research. One potential direction is the further exploration of the GGEV distribution in the context of various hydrological and meteorological phenomena. Given its superior performance in fitting most samples and its sensitivity to trends, especially under non-stationary conditions such as climate change, future studies could examine its applicability across different geographical regions and climatic conditions.

The findings on the influence of catchment types on distribution parameters indicate that more research is needed to refine our understanding of how landscape characteristics interact with hydrological distributions. A deeper exploration into the relationship between the catchment area characteristics, especially in varied topographies and land-use patterns, could yield more universal insights. Expanding the range of predictor variables used in modeling, beyond trend detection, nature of catchment, catchment area, and the hydrological characteristics, might also improve the accuracy and flexibility of distribution selection.

As part of future research directions, the team plans to focus on modeling the GGEV distribution in a full Bayesian approach, involving the estimation of the four parameters of this distribution using MCMC sampling, instead of the ggevp function. In particular, it is planned to determine prior distributions for each of the parameters and to implement the MCMC algorithms, taking into account the need to adjust the proposal distributions, the length of the burn-in phase, thinning, and the number of samples — depending on the specifics of the model under study."

**Technical corrections:**

1. Line 355: "the widest range of area A is characterized by samples fitted to the GGEV distribution (35–1,500 km²)", according to Figure 8, it seems that the GEV distribution actually covers the widest range of catchment area.

**Response:** We thank the **Referee** for pointing this out. The paragraph has been revised in lines 389–396, with the changes marked in red. Below is the updated version exactly as it appears in the manuscript:

"For the PT samples (Figure 8, Table S5), the widest range of area A is characterized by samples fitted to the GEV distribution (48-2,470 km²), while other distributions (GGEV, LN3, P3) have more limited ranges. The widest range of Qp is characterized by samples fitted to the GGEV distribution (8-750 m³/s), indicating its flexibility in modeling extreme flows. The P3 and LN3 distributions have narrower ranges, making them less flexible for samples with a detected PT. In another study on the evaluation of the GEV and LN3 distributions, carried out for the Sewden River, Kousar et al. (2020), using the L-moments estimation, concluded that two locations, one with A in the range of 1000-10,000 km², and one with A above 10,000 km², exhibit a platykurtic distribution and fit best to the GEV. In turn, the LN3 distribution was the best fit for three other locations that exhibit a leptokurtic distribution for areas ranging 100-1000 km², 1000-10,000 km², and above 10,000 km²."

2. Line 467: "In the second RDA, the first two axes (RDA 1 and RDA 2) explain 54.36% of the variance (63.60% and 22.86%, respectively)", the correct total should be 86.46%, not 54.36%.

**Response:** We thank the **Referee** for bringing this to our attention. The total explained variance has been corrected in the manuscript. The full revised text is included under section 2: Specific Comments. Below is the sentence fragment with the corrected value from the revised manuscript:

"In the second RDA, they explain 86.46% (63.60% and 22.86%, Fig. 9b),..."

**Referee 2**

This manuscript evaluates the applicability of the Dual Gamma Generalized Extreme Value (GGEV) distribution for flood frequency analysis using data from a large set of catchments in Poland. While the study addresses a relevant topic and applies a technically interesting distribution, major revisions are needed before publication—particularly regarding the inconsistent use of estimation methods (Bayesian vs. MLE), limited justification for the choice of GGEV, excessive detail in the redundancy analysis, and recurring issues with redundancy, figure quality, and sentence clarity, among others. Detailed comments are provided below.

**Mayor comments**

1) The current title, "The application of new distribution in determining extreme hydrologic events such as floods", is somewhat vague and potentially misleading, as it

implies that the distribution was developed or derived as part of this study. However, the manuscript focuses on applying an existing distribution—the Dual Gamma Generalized Extreme Value Distribution (GGEV)—to flood frequency analysis.

I suggest revising the title to name the distribution, improving clarity and precision explicitly. For example, "Application of the Dual Gamma Generalized Extreme Value Distribution to Extreme Hydrologic Events" would more accurately reflect the scope and contribution of the paper.

**Response:** Thank you to the Referee for pointing this out. We agree with the suggestion. Since the research concerns the territory of Poland, which lies within the temperate climate zone, we decided to include this information in the title. The title "The application of new distribution in determining extreme hydrologic events such as floods" has been replaced with "Evaluation of the Dual Gamma Generalized Extreme Value Distribution for flood events in Poland".

2) The manuscript adopts the GGEV distribution as a flexible alternative to traditional three-parameter models for flood frequency analysis under non-stationary conditions. However, the rationale for selecting this distribution and the hydrologic interpretation of its parameters—particularly the additional shape parameter (delta)—could be more clearly articulated.

It would strengthen the manuscript if the authors provided a more detailed explanation of how the GGEV distribution improves the modeling of extreme events, for example, by showing how delta influences tail behavior or skewness in a hydrologically meaningful way. Comparative visualizations (e.g., PDFs or return level plots) could also help clarify the added value of this distribution in the context of practical flood risk applications.

**Response:** We thank the Referee for this comment. During the course of our research, we indeed created the types of plots mentioned by the Referee in order to investigate how the additional shape parameter in the GGEV distribution improves the modeling of extreme events – particularly, how it affects the shape of the extreme value distribution. To address this, we have added three plots, which are included in the manuscript as Figure 13a–c, along with a corresponding caption. This addition has been inserted at line 565 of the manuscript:

Fig. 13. Empirical (black dots) and theoretical probability density functions of GEV and GGEV for Qp, shown for three profiles: (a) 149180130, (b) 149180230, (c) 149190010.

"In order to investigate how the  $\delta$  parameter in the GGEV distribution affects the form of the extreme value distribution, a visual inspection was conducted (Fig. 13). Figure 13 shows the ability of both GEV and GGEV distributions to model hydrological extremes. At profile 149220070, the GGEV captures the peak of the distribution more accurately than GEV. For station 150190080, the GGEV captures the peak of the distribution more accurately and GGEV provides a slightly better fit in the upper tail. At station 150200090, both models follow the empirical curve well, although GGEV shows a marginal advantage in modeling the higher extremes."

3) The study employs Bayesian MCMC estimation for the GGEV parameters, while using maximum likelihood estimation (MLE) for the GEV, LN3, and P3 distributions. This inconsistency introduces potential biases in model comparison and undermines the objectivity of the goodness-of-fit evaluation.

It is strongly recommended that the authors either apply a uniform estimation framework across all distributions (e.g., MLE or Bayesian) or explicitly justify using Bayesian inference for the GGEV distribution alone. If the Bayesian approach is retained for GGEV, it would be important to include appropriate posterior diagnostics (e.g., trace plots, convergence statistics) and sensitivity analyses regarding prior assumptions.

Additionally, in the context of Bayesian modeling, it may be more appropriate to incorporate probabilistic performance metrics that reflect both accuracy and reliability —such as the Continuous Ranked Probability Score (CRPS) or its skill score version (CRPSS) (Hersbach, 2000; Ossandón et al., 2022)— rather than relying solely on point-estimate-based metrics like MAE or RMSE. This would provide a more robust basis for comparing predictive performance across models.

**Response:** The Referee's concerns are valid because MLE provides only a point estimate of the fitted model parameters (frequentist theory treats parameters as fixed values with an associated random error), the MCMC recovers the entire posterior distribution of the model parameters given in the data, providing additional information such as parameter uncertainty and correlations (Bayesian analysis treats all the parameters as random values) (Hamra et al., 2013; Harms and Roebroeck, 2018). MCMC sampling requires adjustment and tuning, specific to each model, particularly in the choice of proposal distributions, burn-in length, thinning, and the number of samples to store (Harms and Roebroeck, 2018). However, the MCMC we used is not entirely a full Bayesian approach – we are not applying the aforementioned full procedure according to the instructions of the distribution's creator (Nascimento et al., 2015) and packages MCMC4Extremes (Silva and Do Nascimento, 2022). In line 175, we stated that the GGEV distribution parameters are estimated using the MCMC method, and the value of the additional shape parameter is selected after determining its optimal value using the Akaike Information Criterion (AIC) according to the articles (Nascimento et al., 2015; Silva and Do Nascimento, 2022). In fact, three parameters: location parameter, scale parameter, shape parameter are estimated a posteriori by the ggevp function from the MCMC4Extremes package (in this study: block = 1, int = 100; computer: AMD Ryzen 7 4800H with Radeon Graphics 2.90 GHz, RAM: 16GB; SSD: 500 GB and 1TB). The first thin\*int/3 iterations is used as burn-in. The Metropolis-Hastings algorithm technique of sampling was used. However, the fourth parameter - the additional shape in the MCMC estimation is constant because identifiability problems were detected in the estimation of the parameters as stated by (Nascimento et al., 2015). In this case, Nascimento et al. (2015) created a grid of possible values for the additional parameter to estimate the other parameters using the Bayesian approach for each point of the grid, and choose the one grid point that has the lowest -2ln(L) to calculate AIC, BIC and DIC. Therefore, the value of the fourth parameter (add shape) is not estimated directly from the posterior distribution. It can be assumed that this is not a purely Bayesian approach, but limited to the capabilities of the ggevp function. It can also be assumed that this approach is a certain compromise between a full Bayesian estimation and an approach more similar to the a priori model, where the parameter is treated as fixed rather than random.

The code in R, as described by (Silva and Do Nascimento, 2022), and the code from this article is given below:

R > w = rggev(2000, 0.4, 10, 5, 0.5)

```
R > delta=seq(0.1,2,0.2)
R > results=array(0,c(length(delta),4))
R > for (i in 1:length(delta)) {
R+ ajust=ggevp(w,1,1000,delta[i])
R+ results[i,]=summary(ajust)$fitm}
R > resultsb=cbind(delta,results)
R > colnames(resultsb)L'c("delta","AIC","BIC","pD","DIC")
R > resultsb
```

The resultsb object shows a table with the fit measures of estimation for each value of add shape (delta,  $\delta$ ). In our study, add shape (delta,  $\delta$ ) is considered in a sequence from 0.01 to 10, with an increment of 0.01. Additionally, we slightly modified this code, simultaneously obtaining the mean posteriori (the postmean object). The postmean object is a summary statistic of the GGEV distribution. Markov chains were returned by the \$posterior object, which was used in the gevp function (although in the documentation of the MCMC4Extremes package; the function is used to estimate the parameters of the GEV distribution, taking into account the Bayesian approach); this possibility is confirmed by scientific work (Belzile et al., 2023). However, we did not perform further analysis of the Markov chains before the review. Additionally, as stated by (Silva and Do Nascimento, 2022), the use of the above-cited code from the ggevp function, which we also used, refers to priors that are non-informative. The package automatically applies general, wide priors (e.g., flat, i.e., those that minimally affect the posterior result). However, the Bayesian approach (not used in this study) usually allows for a choice between informative and non-informative priors, e.g. the evdbayes package. In turn, the advantage of the ggevp function we used lies in its simplicity – this function works without the need to program the entire model structure. The choice of priors is fixed, and the new GGEV extension is dedicated to hydrological extreme events. This package (MCMC4Extremes) includes, among others, the option of automatically determining posterior means (postmean), credibility intervals (postCI) as well as the predictive distribution without the need for further MCMC chain analysis.

In summary, the used posterior distribution has an additional parameter not determined by the MCMC method, the choice of priors is fixed, and the priors are not informative, which greatly limited the possibilities and the Bayesian approach, which is why we believe that the GGEV model (MCMC+grid of values) and the three-parameter models (MLE) can be compared. Therefore, and after a thorough analysis of our manuscript, we believe that we should thank the Referee for pointing out the very important fact of comparing different models whose parameters are estimated by different models. We believe that adding the above arguments to the article will resolve the doubts about the methods we have adopted.

For greater clarity of the manuscript, we changed subsection "3.3 Extreme value distributions" to "3.3 Extreme value distributions and parameter calibration methods".

In the article, in lines 175 to 187, the content will be updated as follows:

"This GGEV is a four-parameter extension of the (GEV) distribution with an additional shape parameter  $(\delta)$ . This GGEV is a four-parameter extension of the (GEV) distribution with an additional shape parameter  $(\delta)$ . The three parameters of this distribution:  $\mu$ ,  $\sigma$ ,  $\xi$  were estimated a posteriori by the ggevp function from the MCMC4Extremes package (in this study: block = 1, int = 100; computer: AMD Ryzen 7 4800H with Radeon Graphics 2.90 GHz, RAM: 16GB; SSD: 500 GB and 1TB). The initial 33,333.3 iterations - corresponding to the first portion of the chain (first thin\*int/3 iterations) - were designated as the burn-in period. The Metropolis-Hastings algorithm

technique of sampling was used to estimate the marginal posterior distribution for each parameter, except for the  $\delta$  parameter because identifiability problems were detected in the estimation of this fourth parameters as reported by (Nascimento et al., 2015). For the estimation of the  $\delta$  parameter, the method proposed by Nascimento et al. (2015) and (Silva and Do Nascimento, 2022) was adopted. We created a grid of possible  $\delta$  values from 0.01 to 10, with an increment of 0.01, to estimate the other parameters ( $\mu$ ,  $\sigma$ ,  $\xi$ ) using the Bayesian approach for each point of the grid, and selected the  $\delta$  value with the lowest -2ln(L). The calibration of the fourth parameter  $\delta$  is thus not based on a purely Bayesian approach, but limited to the capabilities of the ggevp function (Silva and Do Nascimento, 2022).

The posteriori parameter values were selected by the lowest AIC, BIC and DIC values (Belzile et al., 2023)."

We agree with the referee that the choice of MCMC over MLE or vice versa should be justified. It is worth mentioning here that although the creators of the GGEV distribution provided formulas for the MLE, they also conducted studies showing that using the MCMC methods efficiently recovers the true parameter values of the generalizations. The credibility intervals obtained through these methods demonstrated high accuracy in relation to the true parameter estimates (Nascimento et al., 2015; Silva and Do Nascimento, 2022). In turn, our previous studies with three-parameter distributions and the studies of other scientists have shown that for the a priori estimation of distributions, MLE fitted theoretical curves well to the actual values. Examples can be found in the published studies (Gruss et al., 2020, 2022).

However, GGEV has 4 parameters, hence the MLE analysis is more difficult to perform compared to three-parameter distributions, and the additional parameter, similar to the posterior distribution, was determined using a grid of values. Furthermore, we believe that parameter estimation using MLE and a grid of values can be time-consuming, but this needs to be verified and may be a topic for further research. The use of MLE for estimating the GGEV distribution has a drawback: it can give incorrect results (we will receive an NA message) when any of the data point exceeds the theoretical boundary of the distribution for a given set of parameters (1).

$$1 + \xi \frac{(x_i - \mu)}{\sigma} > 0, dla \ i = 1, \dots, n$$

where:

 $\mu$  – location parameter

 $\sigma$  – scale parameter

 $\xi$  – shape parameter

 $\delta$  – shape parameter of the GGEV extension.

(1)

In turn, parameter estimation by MCMC has the same drawback because the posterior probability distribution also contains a component resulting from the likelihood function, which contains the same condition. However, the ggevp function allows for the rejection of such a set of parameters, in which case another set is drawn.

It is also worth mentioning that methods based on the Bayesian theory, such as the MCMC methods, offer a fundamentally different approach than those based on the maximum likelihood to obtain the distribution of interesting parameters. Furthermore, from a theoretical point of view, these two procedures try to solve different problems. The Frequentist theory deals with the long-term properties (bias, mean squared error) of estimators with repeated sampling, while the Bayesian theory deals with calculating the posterior distribution from a single sample of analysis and the prior distribution (Hamra et al., 2013). Both methods have its limitations. The MLE method performs poorly with small samples and the MCMC method has its problems of

bias (confounding, information, and selection) and hardware requirements that can limit computational performance. However, in the absence of an informative prior distribution, these two techniques will often return similar, if not identical, point and interval estimates for the parameters of interest. In some model specifications, MCMC approaches and asymptotic maximum likelihood approaches can give similar results for moderately sized samples (Hamra et al., 2013). However, there are works such as Ng et al. (2024), that indicate opposite results, where MLE is better than MCMC, or similar effects occur in parameter estimation, such as those in the research of Yılmaz et al. (2021). A method similar to ours was used by Kyojo et al. (2024) comparing the GEV distribution, whose parameters are estimated by the classical MLE method and the Bayesian MCMC by including non-informative a priori data. Comparisons between the results concerning the Bayesian approach (TCEV) and the classical approach (LP3-PLIF) in the discussion of the results were also used by Totaro et al. (2024). Other results regarding the comparison of the classical method and the Bayesian method are presented in the works Shehata et al. (2024). This also led us to compare the 3-parameter distributions with a limited MCMC estimation of a 4-parameter distribution.

Moving forward to ensuring objectivity, we propose to include information on future research directions at the end, after the conclusions, where we will state that the team plans further research on parameter estimation using the MCMC method, not only with the ggevp function from the MCMC4Extremes package but also with a Bayesian approach, assuming that MCMC sampling requires adjustment and tuning, specific to each model, particularly in the choice of proposal distributions, burn-in length, thinning, and the number of samples to store. Therefore, we propose adding the following to line 656:

"As part of future research directions, the team plans to focus on modeling the GGEV distribution in a full Bayesian approach, involving the estimation of the four parameters of this distribution using MCMC sampling, instead of the ggevp function. In particular, it is planned to determine prior distributions for each of the parameters and to implement the MCMC algorithms, taking into account the need to adjust the proposal distributions, the length of the burn-in phase, thinning, and the number of samples — depending on the specifics of the model under study."

We thank the referee for pointing out the need to add appropriate a posteriori diagnostics of the GGEV distribution and sensitivity analyses regarding prior assumptions. We agree with the Referee that the user must determine whether inference from their MCMC samples is trustworthy. This involves evaluating whether each chain appears to be stationary and assessing the quality of its mixing (Magee et al., 2024). Since only 3 parameters of the GGEV distribution  $(\mu, \sigma, \text{ and } \xi)$  are estimated a posteriori by the ggevp function from the MCMC4Extremes package, a posteriori diagnostics and sensitivity analysis were performed for them.

We created trace plots for a single chain and for three chains. We used the plot.mcmc function from the coda package for this. We made trace plots separately for each of the three parameters ( $\mu$ ,  $\sigma$ , and  $\xi$ ). For the  $\mu$  parameter in each profile (in 678 profiles), the chain reached stationarity (the  $\mu$  values oscillate around the mean for most of the iterations). For the  $\xi$  parameter in eight profiles, and for  $\sigma$  in 58 profiles, the chain did not reach stationarity. When analyzing the plot with three chains, it can be seen that their courses are quite consistent for all of the three analyzed parameters (they overlap and do not separate clearly from each other, which suggests their convergence).

In addition, to assess the convergence of the model, the scale reduction factor  $\hat{R}$  (Gelman and Rubin's convergence diagnostic) was run for three parallel chains (Hamra et al., 2013; Ossandón et al., 2022). For this purpose, the gelman.diag function from the coda package was used.  $\hat{R}$  values below the critical threshold of 1.1 indicate adequate model convergence. In all our runs, R values were below 1.1 in 678 samples, confirming convergence. The  $\hat{R}$  factor is commonly

seen as a convergence diagnostic, useful for finding sufficient burn-in (Jones and Qin, 2022; Vats and Knudson, 2021).

In the manuscript, on line 188, after the above-mentioned additions, we will add:

"The diagnosis was for each chain based on trace plots using the plot.mcmc function from the coda R package (Plummer et al., 1999). We made trace plots separately for each of the three parameters ( $\mu$ ,  $\sigma$ , and  $\xi$ ). For the  $\mu$  parameter, in each profile (in 678 profiles), the chain reached stationarity (the values of  $\mu$  oscillate around the mean for most of the iterations). For the  $\xi$  parameter in 8 profiles, and for  $\sigma$  in 58 profiles, the chain did not reach stationarity. When analyzing the plot with three chains, it can be seen that their courses are quite consistent for all of the three analyzed parameters (they overlap and do not separate clearly from each other, which suggests their convergence).

Moreover, to assess the convergence of the model, the scale reduction factor  $\hat{R}$  (Gelman and Rubin's convergence diagnostic) was run for three parallel chains (Hamra et al., 2013; Ossandón et al., 2022). For this purpose, the gelman.diag function from the coda R package was used. The  $\hat{R}$  values below the critical threshold of 1.1 indicate adequate model convergence. In all our runs, the  $\hat{R}$  values were below 1.1 in 678 samples, confirming convergence. The  $\hat{R}$  factor is commonly seen as a convergence diagnostic, useful for finding sufficient burn-in (Jones and Qin, 2022; Vats and Knudson, 2021)."

We performed a sensitivity analysis as follows: we tried randomly sampling from a best candidate from two-parameter and three-parameter and four-parameter distribution separately and then tested whether these distributions would not "overtune" and lead to more erroneous extrapolations beyond the range of the empirical data used for the calibration. Therefore, we created QQ plots with random quantiles on the Y axis and theoretical quantiles on the X axis. The analysis indicated that the GGEV distribution showed a good fit, the GEV distribution was characterized by a slightly worse fit. Unfortunately, the LN2 distribution did not correctly fit the random and theoretical data. On the workflow (Figure 3), we added an additional point of sensitivity analysis. In addition, on line 288, at the end of the sentence, we will add:

"Moreover, sensitivity analysis was also performed by randomly sampling from a two-, three- and four-parameter distribution separately and then testing whether these distributions would not "overturn" and lead to more erroneous extrapolations beyond the range of the empirical data used for calibration. This evaluation was based on quantile-quantile (QQ) plots, in which the quantiles obtained from the calibrated distributions were compared with the empirical ones."

The following results will be added at the end of line 596: "This is confirmed in the QQ plots; see the Supplementary Material."

We agree with the referee that CRPS provided a more robust basis for comparing predictive performance across the different models analyzed. The Continuous Ranked Probability Score (CRPS) was used to evaluate the cumulative distribution functions. CRPS was used to measure the difference between the empirical CDF and the theoretical CDFs (Bröcker, 2012; Gneiting and Katzfuss, 2014). For this purpose, the crps\_sample function from the scoringRules package was used. Although empirical quantiles and theoretical quantiles were used, the same results were obtained as with the MAE and RMSE tests. Therefore, to improve the article, we will modify our diagram (Figure 3)

Figure 3. Workflow for evaluating overparameterization and overfitting in multi-parameter probability distributions.

and the beginning of 3.4:

**"3.4 Accuracy measures and scoring rules**

The goodness of fit of the four probability distributions to the empirical data was evaluated based on the accuracy measures Mean Absolute Error (MAE) and Root mean square error (RMSE). The MAE is recommended for leptokurtic distributions (MAE) and RMSE is preferred for platykurtic distributions (Karunasingha, 2022). Among the 678 samples, the kurtosis value exceeded 3 for 560 samples (leptokurtic distributions), while kurtosis less than 3 was observed in 118 samples (platykurtic distributions). Moreover, the Continuous Ranked Probability Score (CRPS) was used to compare the entire CDF distribution (Hersbach, 2000; Pic et al., 2025). For this purpose, the crps\_sample function from the scoringRules R package was used (Jordan et al., 2016)."


If the authors choose not to implement non-stationary models (e.g., distributions with time-varying parameters such as mu(t) or sigma(t)), it would be important to include a discussion justifying this decision. Clarifying whether the approach is intentionally based on stratified modeling (e.g., trend-based grouping) rather than a fully process-based framework would help readers understand the scope and limitations of the results, particularly in terms of their applicability to future climate-influenced extremes.

Response: We thank the Referee for this suggestion. We agree that it is an important aspect that, in our opinion, deserves a separate and in-depth study. We have included a note on this point at the end of the conclusions section, in lines 647–649. The aim of our study was to demonstrate how the analyzed distributions behave under various trend conditions, including both the presence and absence of trends, and how distribution parameters respond to these trends. To this end, we applied the well-established Mann-Kendall test to identify monotonic trends. The identified trend was also used as a variable in the redundancy analysis (RDA). In our view, this does not constitute a methodological gap, as it was not within the intended scope of our research. Nevertheless, following the Referee's valuable suggestion, we have added a note about this limitation at the end of the line 619:

" ... ; moreover, the study did not account for the non-stationarity in the parameters of the analyzed distributions.distributions"

5) While the RDA section is methodologically comprehensive, it currently presents an excessive level of detail that overwhelms the reader and hinders the extraction of key insights. The narrative includes lengthy descriptions of biplots and correlation patterns for each distribution, much of which could be streamlined.

I suggest briefly introducing how to interpret RDA results in the Methods section (3.5) and then making the Results section (4) more concise and targeted, focusing only on the most relevant and interpretable patterns. Analyzing each RDA result in detail within the main text is unnecessary. For example, the detailed plots and discussion related to the LN3 and P3 distributions could be moved to the Supplementary Material, especially since the primary focus of this study is the application of the GGEV distribution in flood frequency analysis. This would allow the main text to highlight the most critical findings while avoiding redundancy and improving readability.

**Response:** We thank the Referee for this comment. We agree with it. As suggested by the Referee, an introduction to the interpretation of the RDA results has been added in section 3.5.

**"3.5 Redundancy analysis**

Redundancy analysis (RDA) was applied as a canonical technique, to investigate the influence of environmental variables and sample characteristics on the parameters of the extreme value distributions. The aim was to identify common patterns and key factors affecting the distribution parameters.

The environmental factors examined included the watershed area, categorized by the catchment type, and the nature of the watercourse (Lowlands, Highlands, Mountains) (Bertola et al., 2020; Han et al., 2023; Tyralis et al., 2019). Sample characteristics considered included the highest Qp, MAF, sample size (N), empirical moments of standard deviation (SD), variance (Var), skewness (Skew.), kurtosis (Kurt.), third-moment center (3thMoment), fourth-moment center (4thMoment; which measures the intensity of the distribution tails) of the Qp, and trend measures.

RDA was performed separately for each distribution. The final RDA model was selected by evaluating independent variables using the Variance Inflation Factor (VIF). The independent variables of the RDA are catchment area ranges: a – micro-catchments, b – meso-catchments, c – macro-catchments, d – large catchments, e – very large catchments; Qp; trend (NMT is no trend, PT is positive trend and NT is negative trend); the nature of the watercourse (L – lowlands, H – highlands, M – mountains); Skew., 4thMoment; Kurt., and N. The response variables of the RDA are  $\delta$ ,  $\mu$ ,  $\sigma$  and  $\xi$ .

Since the mean, SD, Var, 3thMoment, and 4thMoment are interrelated, it is essential to carefully select the set of explanatory variables. It was also confirmed that multicollinearity exists between Skew. and Kurt. as explanatory variables (VIF > 10). Collinearity between Skew. and Kurt. may result from the fact that both of these measures are defined using the SD. Therefore, RDA was conducted separately for Kurt. (Fig. 9a, 10a, 11a, 12a) and Skew. (Fig. 9b, 10b, 11b, 12b). Additionally, RDA was performed with the inclusion of the catchment area ranges and Kurt. was replaced with 4thMoment and Skew. (see Figures 9c, 10c, 11c, and 12c).

The decision to replace Kurt. with 4thMoment was made because both Skew. and Kurt. are functions of SD, making them potentially collinear. The use of Skew. and 4thMoment allows for capturing more detailed aspects of the data distribution. 4thMoment measures the overall Kurt., which is the tail heaviness of the distribution, while Kurt. is the normalized version of this moment. Following the initial RDA, subsequent analyses considered only the changes that were not identified in the first analysis.

The use of topography in modeling Qp helps to uncover the runoff mechanism prevailing in the catchment (Valeo and Rasmussen, 2000).

RDA, standardized by response variables (center and standardize) and environmental variables (center), was performed using the Canoco software ver. 5.12 (ter Braak and Šmilauer, 2019)."

In the Results and discussion section, we focused only on the most significant and interpretable patterns. The details regarding the RDA for parameters of other distributions have been removed from the Supplementary Material. The revised text from the manucript is as follows:

[revised manuscript text omitted]
 H (Fig. 11a-b). The M, NT, PT, and NMT are correlated with RDA 3 (Fig. 11a). In the second

RDA,  $\xi$  is weakly negatively correlated with M and PT. (Fig. 11b).  $\mu$  is weakly correlated with NMT and NT (Fig. 11b). In the third RDA,  $\xi$  is inversely proportional to PT (Fig. 11c). The b parameter has a weak positive relation to the  $\xi$  parameter (Fig. 11c). In catchments b,  $\xi$  increases. In contrast, H, L, M, NT, and NMT, c, d do not influence the distribution parameters (Fig. 11c).

**4.4.4 The LN3 distribution**

Fig. 12. RDA results of the relationship between environmental factors, sample characteristics and the parameters of LN3 ( $\sigma$ ,  $\xi$ ,  $\mu$ ), distinguishing between: (a) Kurtosis instead of skewness, (b) Skewness instead of kurtosis, (c) Catchment area ranges instead of variable A and the fourth moment. Descriptions of symbols:  $\mu$  – location parameter,  $\sigma$  – scale parameter,  $\xi$  – shape parameter, catchment area ranges (in km²): a - A < 10;  $b - 10 \le A < 100$ ;  $c - 100 \le A < 1,000$ ;  $d - 1,000 \le A < 10,000$ ; e - A > 10,000; flow peak ( $Q_p$ ); trend (NMT is no trend; PT is positive trend and NT is negative trend); nature of the watercourse (L – lowlands, H – highlands, M – mountains), parameters of the empirical sample (Skew. is empirical skewness, Kurt. is empirical kurtosis and 4thMoment is the  $4^{th}$  center moment); N – sample size.

The first two axes (RDA 1 and RDA 2) explain 46.61% of the variance (38.71% and 7.90%, respectively) (Fig. 12a). In the second RDA, they explain 54.09% (38.98% and 15.11%, respectively) (Fig. 12b), and in the third – 57.72% (42.60% and 15.12%, respectively) (Fig. 12c). A comparison of the RDA results for the LN3 distribution parameters with those of GEV, P3, and GGEV has revealed the following dependencies:  $\mu$  is weakly correlated with NMT (Fig. 12a-b). PT, NT, M, and L are correlated with RDA 3 (Fig. 12a-b). In the first RDA,  $\xi$  is negatively related to Kurt. and H is weakly correlated with N. (Fig. 12a). In the second RDA,  $\xi$  is negatively related to Skew. and H (Fig. 12b-c).  $\mu$  is weakly inversely proportional to N (Fig. 12b). In the third RDA, in contrast to GEV, GGEV, and P3,  $\sigma$  is strongly positively correlated with e and  $\mu$  is strongly positively correlated with Qp and 4thMoment (Fig. 12c).  $\xi$  is positively correlated with NMT. Moreover,  $\xi$  is negatively correlated with N and NT (Fig. 12c).  $\sigma$  is negatively correlated with c (Fig. 12c). As reported (Kamal et al., 2017), the larger the N, the better the result for LN3. Lastly, b, d, H, M, L, and PT are correlated with RDA 3 (Fig. 12c).

**4.4.5 Key points on the influence of environmental factors and sample characteristics**

The following points summarize key findings regarding the relationships between environmental factors and sample characteristics and the parameters of the studied probability distributions:

- GGEV tends to have the H, M, L located outside the RDA1 and RDA2.
- GGEV and P3 share a common feature of a negative correlation between N and  $\xi$ , while GEV and LN3 exhibit more complex correlation patterns.
- GGEV, GEV, and P3 show similar correlations between the 4thMoment and Qp, as well as e in the context of RDA1, whereas LN3 differs in this respect.
- Both GGEV and GEV exhibit a pattern characterized by a negative correlation between NT and  $\xi$ , and a positive correlation between  $\xi$  and NMT.
- Only for the P3 distribution is the  $\xi$  parameter more strongly negatively correlated with N than with Kurt. and Skew. This confirms that the  $\xi$  parameter is highly dependent on N.
- Only the patterns for GGEV and LN3 show a PT along the RDA3.
- Catchment size types influence the distribution parameters, with the most types affecting the parameters of GEV and the least types affecting the parameters of GGEV, LN3 and P3."

Based on the above comparison, the GGEV distribution shows some similarities with other distributions regarding the occurrence and correlation of the distribution parameters. However, there are differences in certain aspects, such as the distribution of parameters in the principal components and parameter correlations, which indicates unique characteristics of GGEV compared to GEV, LN3, and P3. GGEV often differs from other distributions in how its parameters spread within the principal component space, which may be significant when modeling and interpreting the results of extreme flow data analysis."

6) The manuscript has recurring issues related to redundancy and incomplete or unclear sentence structure. In several instances, information is repeated across consecutive lines with minimal added value, affecting the text's overall conciseness and flow. Additionally, some sentences appear grammatically incomplete or are phrased in a way that makes their interpretation ambiguous. These issues may hinder reader comprehension and should be systematically addressed.

For specific illustrations of these concerns, please refer to the minor comments. A thorough language and structural revision is recommended to enhance the manuscript's clarity and readability.

**Response**: We thank the Referee for pointing this out. We have carefully reviewed the manuscript and corrected the stylistic and grammatical errors. Additionally, all variables were formatted in italics in both the manuscript and the supplementary material. The distribution parameters were replaced with Greek symbols, as defined in the methods section, to ensure consistency between the manuscript and the supplementary material. Furthermore, the distribution parameters displayed in the RDA plots were also updated to reflect these symbols. In our opinion, these changes have made the article more consistent, clearer, and easier to understand. The revised text, including our changes and additions, has also been submitted to a native speaker for proofreading.

7) The overall quality and resolution of the figures should be improved to enhance their visual clarity and readability. For instance, in Figure 10, it is difficult to clearly discern the horizontal and vertical lines corresponding to zero on each axis. The axis tick labels, as well as any other textual elements within the figure, appear blurry and are challenging to read. This issue applies not only to Figure 10 but also affects several other figures throughout the manuscript. Enhancing the resolution and font clarity would significantly improve the interpretability of the graphical results and ensure they meet publication standards.

**Response**: Thank you for pointing it out. Figures 3 through 12 have been revised in accordance with the Referee's suggestions as well as the journal's Guide for Authors (Submission, Figures and Tables). The updated figures are in the PNG format, have a resolution of 300 dpi, and do not exceed 5MB in size. Additionally, we used the online tool Coblis – Color Blindness Simulator to ensure that the figures are CVD-friendly.

**Specific comments:**

1) L48: Lognormal is a two-parameter distribution, or do you mean LN3?. Please correct it if it corresponds.

**Response:** We thank the Referee for his attention to detail. Although the section "Abbreviations and acronyms in alphabetical order" explains that LN3 refers to the three-parameter log-normal distribution, we began this sentence of the abstract with: " In this paper we compare three-parameter distributions such as the log-normal, ..." To improve clarity, we revised this part of the abstract in lines 46–50:

"Advancing climate change will necessitate the use of new distributions that are more flexible in adapting to trends and other non-stationarities. In this paper we compare three-parameter distributions such as the log-normal, the Generalized Extreme Value (GEV) and the Pearson type III with the Dual Gamma Generalized Extreme Value Distribution (GGEV). The GGEV is a four-parameter extension of the GEV. The comparison is done under different trend conditions and takes into account the differences in the catchment area and peak flow magnitude."

**2) L82: A linear trend means nonstationarity of the time series. Please check it.**

**Response:** We thank the Referee for pointing this out. The cited authors (Richard M. Vogel and Ian Wilson) assumed stationarity of the series, as they stated that anthropogenic influences would have minimal impact on the stationarity of the observational series. However, after reviewing the entire manuscript, we decided to remove this sentence: "The Pearson Type III distribution provides the best fit to both annual minimum and annual average streamflows, assuming the series is stationary but with a linear trend (Vogel and Wilson, 1996)."

**3) L98: Please correct the citation. It should be (Silva and Do Nascimento, 2022).**

**Response:** We thank the Referee for this valuable comment and truly appreciate it. We have carefully reviewed the entire manuscript and corrected the citation in line 98 and line 183. After the revision, the citation now reads as follows: In line 98,,(Silva and Do Nascimento, 2022)"; in line 183: "(Silva and Do Nascimento, 2022)".

**4) L124-125: Please consider improving the phrasing in those lines for clarity.**

**Response:** We thank the Referee for this comment. The sentence has been revised in the manuscript. The original sentence: "The least number of micro-catchments was recorded (for 2 stream gauge profiles), and the highest number of macro-catchments (for 388 stream gauge profiles)." has been replaced with: "The fewest catchments identified were micro-catchments, represented by only 2 stream gauge profiles, while the most numerous were macro-catchments, represented by 388 profiles."

**5) L144: You already mentioned it in L140. Please avoid repetition.**

**Response:** We thank the Referee for pointing this out. However, the error is actually in line 140, as we initially collected 1,070 stream gauge profiles, from which we filtered 678 profiles (≥30 years of data) according to the procedure described in this subsection. Therefore, the sentence in line 140 of the manuscript has been corrected as follows: "For 1070 gauge stations located in the basins of the Vistula, Oder, coastal rivers, Pregola, and Neman, Dniester, Dunajec, the maximum annual flows were collected."

**6) L226: Canoco 5.12 is an R package? Please specify.**

**Response:** We thank the Referee for his attention to detail. Of course, this refers to a software. We have replaced the sentence in the manuscript: "RDA, standardized by response variables (center and standardize) and environmental variables (center), was performed using Canoco 5.12." with the revised version: "RDA, standardized by response variables (center and standardize) and environmental variables (center), was performed using the Canoco software ver. 5.12 (ter Braak and Šmilauer, 2019)." We hope that the revised sentence now eliminates any ambiguity.

**7) Figure 3. It should say "value of the shape." In the same figure, it should be "generating random samples." Please correct them.**

**Response:** We agree with the Referee. Figure 3 has been revised and inserted into the manuscript. The updated version in the manuscript is as follows:

Figure 3. Workflow for evaluating overparameterization and overfitting in multi-parameter probability distributions.

**8) L258: Include the reference for Brunner et al. (2018) in the reference list. Please check it for the rest of the references.**

**Response:** We thank the Referee for pointing this out. The references for (Brunner et al., 2018; Jaiswal et al., 2022) have been included in the reference list. We have carefully reviewed the entire manuscript to ensure that no other references were omitted from the reference list.

**9) L287: Generalized is redundant in "the generalized GGEV". Please remove it.**

**Response:** We thank the Referee for pointing this out. The sentence in the manuscript: "This is consistent with the findings by Nascimento et al., (2016), who found for maximum monthly flow data that the best model was the generalized GGEV model rather than the GEV model." has been replaced with the sentence: "This is consistent with the findings by (Nascimento et al., 2015), who established, for the maximum monthly flow data, that the best model was the GGEV rather than the GEV."

**10) L291: There is an extra comma; please remove it.**

**Response:** We thank the Referee for their careful observation. The redundant comma in the manuscript has been removed.

**11) Figure 6. In the legend, it should be "P3" instead of "3P3"**

**Response:** We thank the Referee for pointing this out. Figure 6 has been revised and inserted into the manuscript.

**12) Lines 324–325 appear to repeat information already stated in the preceding paragraph. Consider revising this section to avoid redundancy and improve the flow of the text.**

**Response:** Dear Referee, the entire paragraph has been carefully reviewed. We agree with you – the information in this paragraph was repetitive. The sentence has been removed from the manuscript.

**13) L336-338: Please revise the phrasing in the sentence. The sentence appears incomplete or abruptly truncated, and the comparison between the distributions could be clarified for better readability.**

**Response:** We agree with the Referee. We have carefully reviewed the entire subsection 4.3 of the manuscript, which has been revised in accordance with comments 13–16. The revised version of subsection 4.3 is presented below:

**"4.3. Goodness of fit results in relation to the catchment size and peak flow**

Next, it was checked whether the relationship between the catchment area (A) and the registered maximum peak flow  $(Q_p)$  could influence the choice of distribution.

Figure 6. Relationship between the catchment area (A) and the peak flow magnitude (Qp) for observational series with no trend and for the four fitting probability distributions (P3, LN3, GEV and GGEV).

The probability distributions determined for the NMT samples show a relationship between A and Qp, represented by a simple regression line (Figure 6, Table S3). The widest range of A is characterized by the samples for which the P3 distribution (30-20,000 km²) and the GEV distribution (3.5-170,000 km²) provide the best fits, while LN3 (35-110,000 km²) and GGEV (50-70,000 km²) fit more limited ranges. Moreover, the widest range of Qp is characterized by the samples for which the P3 (1.9-7,000 m³/s) and GEV (1.6-7,000 m³/s) distributions fit best, suggesting that these distributions are the most flexible in modeling extreme flows for different basin sizes. In contrast, the LN3 (8.5 to 6.500 m³/s) and GGEV (2 to 6,000 m³/s) distributions show a more limited applicability (Figure 6, Table S3). Since this may appear to contradict the results presented in the previous sections, a similar analysis is carried out below, but separately for each trend category.

Figure 7. Relationship between the catchment area (A) and the peak flow magnitude (Qp) for observational series with a negative trend and for the four fitting probability distributions (P3, LN3, GEV and GGEV).

When we focus on the observational series with a NT (Figure 7, Table S4), the widest range of A is characterized by samples fitted to the GGEV distribution (80-180,000 km²), whereas other distributions (P3, LN3, GEV) have more limited ranges. The widest range of Qp is characterized by samples fitted to the GGEV distribution (5-7,000 m³/s), indicating its versatility in modeling extreme flows for samples with a detected NT. The P3 and LN3 distributions have narrower ranges, making them less flexible for samples with a detected NT. This suggests that the GGEV distribution is particularly well-suited for extreme flow events with NT. Moreover, for the NT samples, the GEV distribution fits much better than any of the other three-parameter distributions.

Figure 8. Relationship between the catchment area (A) and the peak flow magnitude (Qp) for observational series with a positive trend and for the four fitting probability distributions (P3, LN3, GEV and GGEV).

For the PT samples (Figure 8, Table S5), the widest range of area A is characterized by samples fitted to the GEV distribution (48-2,470 km²), while other distributions (GGEV, LN3, P3) have more limited ranges. The widest range of Qp is characterized by samples fitted to the GGEV distribution (8-750 m³/s), indicating its flexibility in modeling extreme flows. The P3 and LN3 distributions have narrower ranges, making them less flexible for samples with a detected PT. In another study on the evaluation of the GEV and LN3 distributions, carried out for the Sewden River, Kousar et al. (2020), using the L-moments estimation, concluded that two locations, one with A in the range of 1000-10,000 km², and one with A above 10,000 km², exhibit a platykurtic distribution and fit best to the GEV. In turn, the LN3 distribution was the best fit for three other locations that exhibit a leptokurtic distribution for areas ranging 100-1000 km², 1000-10,000 km², and above 10,000 km²."

**14) Lines 338–340 appear to repeat information already stated in the preceding lines. Consider revising this section to avoid redundancy and improve the flow of the text.**

**Response:** We agree with the Referee. We have carefully reviewed the entire subsection 4.3 of the manuscript, which has been revised in accordance with comments 13–16. The corrected version of subsection 4.3 was already provided above, in response to comment 13.

15) L355-356: it is stated that the widest range of catchment area (A) is represented by samples fitted to the GGEV distribution. However, based on Figure 8, the GEV distribution spans the widest range of A ( $\sim$ 50 -  $\sim$ 2,500 km2). Please revise the description to reflect this more accurately.

**Response:** We thank the Referee for pointing this out. We have carefully reviewed the entire subsection 4.3 of the manuscript, which has been revised in accordance with comments 13–16. The corrected version of subsection 4.3 was already provided above, in response to comment 13.

**16)** L359: the same as **15)**. Redundant.**

**Response:** We thank the Referee for their careful observation. We agree with the Referee. We have carefully reviewed the entire subsection 4.3 of the manuscript, which has been revised in accordance with comments 13–16. The corrected version of subsection 4.3 was already provided above, in response to comment 13.

**17) L370: Define the term "add. Shape."**

**Response:** We agree with the Referee. The symbols used in the formulas in subsection 3.3 have been standardized, and the transferred sentence in the Methods section has been supplemented with: "The response variables of the RDA are  $\delta$ ,  $\mu$ ,  $\sigma$  and  $\xi$ ."

**18) L423: Add an space before (Fig9. c)**

**Response:** We sincerely thank the Referee for the time and effort devoted to reviewing our analyses. We truly appreciate it. The Referee's comment has been taken into account, and subsections 4.4 and 4.5 have been carefully reviewed, revised, and adjusted in accordance with the suggestions.

19) L456-457: The sentence "It exhibits a strong correlation with the scale and location parameters, whereas this relationship is not observed for the shape parameter (Tabari et al., 2021b)" is unclear. It is unclear what "It" refers to or which variable is being described as correlated. Please clarify the sentence and ensure that the subject and its relationship to the distribution parameters are explicitly stated.

**Response:** We sincerely thank the Referee for the time and effort devoted to reviewing our analyses. We truly appreciate it. The Referee's comment has been taken into account, and subsections 4.4 and 4.5 have been carefully reviewed, revised, and adjusted in accordance with the suggestions.

20) Figures 9 and 10: It looks like for GEV, the shape parameter is less sensitive to the sample size (N) that the shape and add. shape for GGEV. If this is the case, it should be highlighted.

**Response:** We sincerely thank the Referee for the time and effort devoted to reviewing our analyses. We truly appreciate it. The Referee's comment has been taken into account, and subsections 4.4 and 4.5 have been carefully reviewed, revised, and adjusted in accordance with the suggestions.

**21) L531: repetition. You already said it in L532.**

**Response:** We sincerely thank the Referee for the time and effort devoted to reviewing our analyses. We truly appreciate it. The Referee's comment has been taken into account, and subsections 4.4 and 4.5 have been carefully reviewed, revised, and adjusted in accordance with the suggestions.

22) L627: In the sentence "the fitted LN2 distribution has a shape parameter value greater than 1 for only seven stations", there appears to be a conceptual error. As far as I understand, the two-parameter log-normal distribution (LN2) includes only a location (mu) and a scale (sigma) parameter and does not have an explicit shape parameter. Please clarify whether this refers to sigma being interpreted as a shape proxy or if there has been confusion with another distribution.

**Response:** We thank the Referee for this important comment. Indeed, the two-parameter lognormal (LN2) distribution is defined by a location parameter ( $\mu$ ) and a scale parameter ( $\sigma$ ), and it does not include a formal shape parameter. In the sentence mentioned, we mistakenly referred to the scale parameter ( $\sigma$ ) as a "shape parameter". This was based on our use of the Lognormal function from the {stats} package in R, where  $\sigma$  (the standard deviation of the log-transformed variable) influences the tail behavior of the distribution. While  $\sigma$  is not a shape parameter in the strict theoretical sense, it was informally interpreted as a shape proxy in this context. We have corrected the terminology in the manuscript to avoid confusion and ensure consistency with the formal definition of LN2. The sentence: "In turn, the fitted LN2 distribution has a shape parameter value greater than 1 for only seven stations." has been replaced in the manuscript with: "In turn, the fitted LN2 distribution has a scale parameter ( $\sigma$ ) greater than 1 for only seven stations, which indirectly affects the distribution's shape."

**Response:** We thank the Referee for the valuable suggestions and the recommended literature. These sources were taken into account during the revision and enhancement of our manuscript.

---

## Author Response (AR2)

Dear Editor,

thank you very much for accepting the article. As requested, I have revised the supplementary material and created a finalized version in PDF format. I also double-checked the references in this file to ensure accuracy. In the manuscript, I updated Figures 6–8 and 13 in accordance with the publisher's guidelines to improve accessibility for readers with color vision deficiencies. Specifically, for Figures 6–8, I adjusted both the colors and the line styles (now using solid, dashed, dotted, and dot-dash lines). For Figure 13, I modified the colors and applied solid and dashed line styles.

Additionally, I realized that I had initially omitted the Financial support section. I have now added this information in lines 672–674. If it is too late to include a separate subsection for this, I would kindly ask that this information be incorporated into the Acknowledgments section instead. Thank you again for your support.

Kind regards,

Łukasz Gruss

(on behalf of all the co-authors)